# Long-term efficient organic photovoltaics based on quaternary bulk heterojunctions

Minwoo Nam[1,*], Minjeong Cha[1,*], Hyun Hwi Lee[2], Kahyun Hur[3], Kyu-Tae Lee[1], Jaehong Yoo[1], Il Ki Han[4], S. Joon Kwon[4] & Doo-Hyun Ko[1]

A major impediment to the commercialization of organic photovoltaics (OPVs) is attaining long-term morphological stability of the bulk heterojunction (BHJ) layer. To secure the stability while pursuing optimized performance, multi-component BHJ-based OPVs have been strategically explored. Here we demonstrate the use of quaternary BHJs (q-BHJs) composed of two conjugated polymer donors and two fullerene acceptors as a novel platform to produce high-efficiency and long-term durable OPVs. A q-BHJ OPV (q-OPV) with an experimentally optimized composition exhibits an enhanced efficiency and extended operational lifetime than does the binary reference OPV. The q-OPV would retain more than 72% of its initial efficiency (for example, 8.42–6.06%) after a 1-year operation at an elevated temperature of 65 °C. This is superior to those of the state-of-the-art BHJ-based OPVs. We attribute the enhanced stability to the significant suppression of domain growth and phase separation between the components via kinetic trapping effect.

[1] Department of Applied Chemistry, Kyung Hee University, Yongin, Gyeonggi 17104, Republic of Korea. [2] Pohang Accelerator Laboratory, Pohang, Gyeongbuk 37673, Republic of Korea. [3] Computational Science Center, Korea Institute of Science and Technology, Seoul 02792, Republic of Korea. [4] Nanophotonics Research Center, Korea Institute of Science and Technology, Seoul 02792, Republic of Korea. * These authors contributed equally to this work. Correspondence and requests for materials should be addressed to S.J.K. (email: cheme@kist.re.kr) or to D.-H.K. (email: dhko@khu.ac.kr).

Bulk heterojunction (BHJ) organic photovoltaics (OPVs) composed of a conjugated polymer donor (D) and a fullerene derivative acceptor (A) have attracted considerable attention in both fundamental and practical areas. Recently, including an additional conjugated polymer donor (or fullerene acceptor) in the binary blends to create ternary OPVs[1] has been advanced as a facile way to broaden the light absorption band, improve the nanomorphology, increase the open-circuit voltage ($V_{OC}$), and boost the charge (or energy) transfer of the OPVs[2–7]. In particular, ternary OPVs based on the efficient non-crystalline polymer host (for example, poly[[4,8-bis[(2-ethylhexyl)oxy] benzo[1,2-b:4,5-b′]dithiophene-2,6-diyl][3-fluoro-2-[(2-ethylhexyl) carbonyl]thieno[3,4-b]thiophenediyl]], PTB7) were shown to exhibit a power conversion efficiency (PCE) of about 10%[6,7], which would meet commercialization requirements.

Besides the demand for high PCE, it is also critical to preserve an optimized morphology of the pristine active layer during the operation of the OPV. The domains of the polymers and fullerenes grow during operation due to their thermodynamic instability at elevated temperatures[8–11], which is a major degradation mechanism limiting the commercial viability of the OPV. In particular, at outdoor operating temperatures, which often exceed the glass transition temperature ($T_g$) of typical polymers[12], both non-crystalline polymers and fullerene derivatives (for example, [6,6]-phenyl $C_{71}$ butyric acid methyl ester, $PC_{71}BM$) are readily clustered accompanying reduction in the D–A interfacial areas and increase in the percolation threshold for charge carrier transport[13]. To extend the lifetime of OPVs, therefore, it is desirable to design an active layer with incorporated additives that effectively alter the thermodynamics of phase separation between the donors and acceptors to slow down the rate of the phase separation[14]. From this standpoint, employing an appropriate D or A additive as a morphology stabilizer would kinetically arrest the morphology at its optimum, enabled by providing parameters to control blending and separation behaviours of components[15]. For instance, the facile technique of incorporating a fullerene additive is advantageous in controlling domain growth in the BHJs by restricting diffusion of low-molecular-weight fullerenes with no requirement of additional chemical modification or post-treatment[15–18]. In addition, ternary OPVs provide adjusted recombination mechanisms compared with their binary counterparts, which would prolong the lifetime of OPVs under degradation conditions[3,6,19]. The main idea behind the method in the present study is to design an active layer employing multifunctional additives that broaden the absorption spectrum, facilitate transfer kinetics and retain optimized morphology.

Here, we demonstrate quaternary BHJ (q-BHJ) blends as a novel platform to achieve highly efficient OPVs that are stable over the long term. The q-BHJ-based OPV with optimized compositions of the donors, PTB7 and poly[N-9′-heptadecanyl-2, 7-carbazole-alt-5,5-(4′,7′-di-2-thienyl-2′,1′,3′-benzothiadiazole)] (PCDTBT), and the acceptors, $PC_{71}BM$ and $PC_{61}BM$, displays enhanced light absorption, an increased exciton dissociation rate, and facilitated charge transport. More importantly, the quaternary OPV (q-OPV) exhibits improved long-term stability at elevated temperatures. Experimental measurements together with simulations reveals that the q-OPV can yield a PCE of more than 6.06% after 1 year at 65 °C, retaining more than 72% of its initial PCE, while the binary reference OPV would suffer from a significantly reduced PCE of 3.39%. Based on the detailed experimental and numerical analysis on the early and long-term performance decay mechanism, we find the improvement in stability to be attributed to impeded domain growth and segregation between polymers and fullerenes in the quaternary blends. We expect the q-BHJ platform to provide a feasible and new approach for achieving long-lasting high-efficiency OPVs that can be used in practical applications.

## Results

**Device structure and optical properties of materials**. To fabricate OPVs, we employed an inverted structure composed of indium tin oxide (ITO)-glass/poly [(9,9-bis(3′-(N,N-dimethylamino) propyl)-2,7-fluorene)-alt-2,7-(9,9-dioctylfluorene)] (PFN)/q-BHJ active layer/MoO$_3$/Ag, as illustrated in Fig. 1a. For the active layer, two conjugated polymer donors and two fullerene derivative acceptors (molecular structures in Fig. 1b) were chosen by carefully considering their absorption profiles and cascade energy levels (Fig. 1c)[20–22]. The q-BHJ composition in this study is advantageous for the fabrication of OPVs due to the simple process used and the general availability of the organic materials employed. The thickness of the active layer was globally fixed at $90 \pm 5$ nm to exclude differences in photovoltaic performance between samples resulting from thickness-dependent effects (Supplementary Fig. 1). As shown in Fig. 1d, the incorporation of PCDTBT in the binary PTB7:$PC_{71}BM$ blend substantially increased the overall spectral response of the active layer toward a broad range of the ultraviolet–visible spectrum. The photoluminescence spectra analysis indicates energy transfer between PCDTBT and PTB7 (Supplementary Note 1, Supplementary Fig. 2). In particular, the absorption band of PTB7 substantially overlapped the emission band of PCDTBT, which in turn enables the efficient energy transfer from PCDTBT to PTB7 (refs 5,23). A further investigation for the detailed charge transfer mechanism driven by the cascade-energy-level have currently been underway. The optical response was even further enhanced in the ultraviolet range below 380 nm when the second acceptor $PC_{61}BM$ was added, ultimately forming the quaternary blend. This optical response implied that the q-OPV was better at harnessing broadband solar energy (ultraviolet –visible–near-infrared (NIR)) than were the binary blends. The optical simulation results based on the T-matrix method agreed well with the experimental spectra (Supplementary Note 2, Supplementary Figs 3a,b). The ability to control the ratio of PTB7 to PCDTBT and that of $PC_{71}BM$ to $PC_{61}BM$ in the q-BHJs (multiple choices of other compositions) allows for the absorption spectrum and hence the colour of the device to be tuned (Supplementary Fig. 4). Such colour tunability is particularly advantageous for aesthetic applications such as colourful building-integrated photovoltaics or luminescent smart windows[24].

**Optimum composition of the quaternary OPV**. To find the optimized composition of the q-OPVs, we began with the best binary D–A ratio of 1:1.5 by weight[20,25] and varied the ratio of donors in the mixture such that PTB7:PCDTBT:$PC_{71}BM = 1 - x$: $x$:1.5, where $0 \leq x \leq 1$. As a good solvent for the reference PTB7:$PC_{71}BM$, chlorobenzene was employed to experimentally optimize the reference device performance (Supplementary Fig. 5). It was more practical to find the optimized composition by varying the relative amounts of the two donors while keeping the amount of the acceptor fixed because this strategy yielded more possibilities for designing the optical and morphological properties of the active layer than did an alternative strategy involving varying the amounts of the acceptors[26]. Figure 2a,b shows the photovoltaic properties for different values of $x$; each value was an average of measurements from more than 16 cells. As apparent from the figures, $0.1 \leq x \leq 0.2$ would be responsible for achieving maximum short-circuit current density ($J_{SC}$) and fill factor (FF). In contrast, $V_{OC}$ slightly increased with increasing $x$. When combined, a maximum PCE was observed when $x = 0.1$.

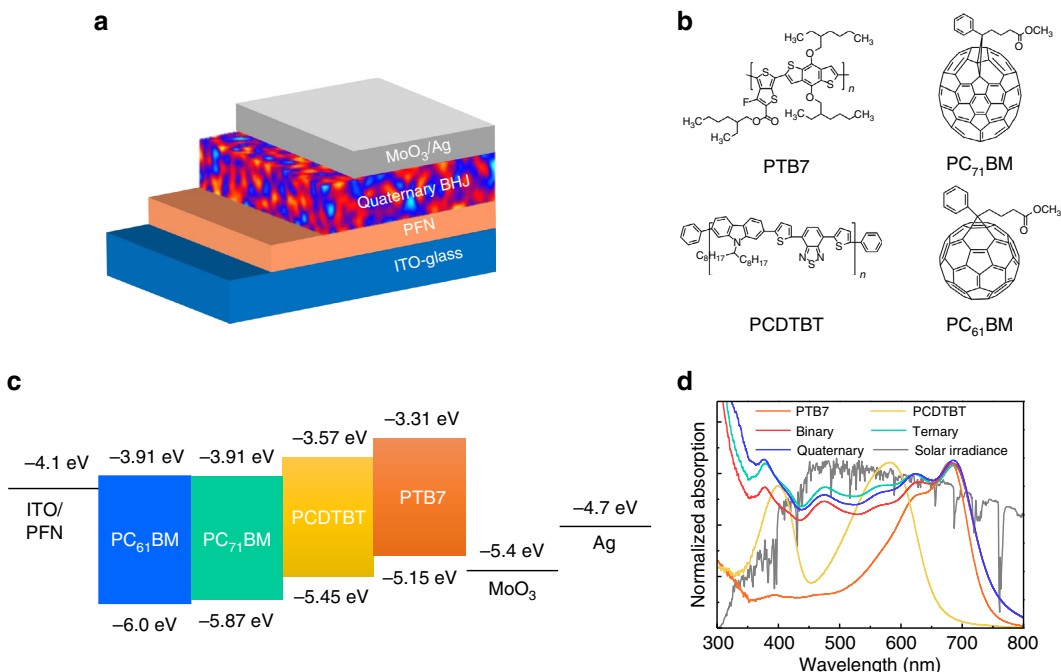

**Figure 1 | Device structure and optical properties of materials.** (**a**) Schematic diagram of the OPV, (**b**) molecular structure of the D and A materials and (**c**) energy band diagram of the q-OPV. The energy levels of PTB7, PCDTBT, $PC_{71}BM$ and $PC_{61}BM$ were obtained from refs 20–22. (**d**) Ultraviolet–visible–NIR absorption spectra of films of pure PTB7, pure PCDTBT, binary PTB7:$PC_{71}BM$ (1.0:1.5) blend, ternary PTB7:PCDTBT:$PC_{71}BM$ (0.9:0.1:1.5) blend and quaternary PTB7:PCDTBT:$PC_{71}BM$:$PC_{61}BM$ (0.9:0.1:1.2:0.3) blend.

Having optimized the composition of the donors, we set out to further improve performance by introducing $PC_{61}BM$ as an additional acceptor and to produce quaternary D–A BHJs. In addition, this approach facilitated controlling the nanoscale morphology of the q-BHJ. It can be conjectured that the additional D and A would interfere with the formation of either crystalline grains of donor polymers or aggregated domains of acceptor fullerenes. Then, the phase separation would be effectively controlled by the morphology to derive long-term and thermal stability, which is demonstrated below in detail. We varied the ratio of the acceptors in the mixture while fixing the overall donor-to-acceptor ratio such that PTB7:PCDTBT: $PC_{71}BM$:$PC_{61}BM = 0.9:0.1:1.5 - y:y$, where $0 \leq y \leq 1.5$. Figure 2c,d compares the dependence of the photovoltaic parameters of q-OPVs on $y$ (mean values out of more than 16 cells). Increasing the $PC_{61}BM$ fraction resulted in higher FF values but lower $J_{SC}$ values. $V_{OC}$ did not vary much as $y$ was changed. Note, in particular, that the enhancement of the FF would be attributed to the balanced charge carrier mobilities or morphologically favourable alteration when the fraction of $PC_{61}BM$ was increased (see hole-to-electron mobility ratio in Supplementary Fig. 6a, Supplementary Table 1)[27]. Details for the mobility test can be found in Supplementary Methods. Ultimately, the PTB7: PCDTBT:$PC_{71}BM$:$PC_{61}BM$ composition yielding the maximum PCE was experimentally found to be 0.9:0.1:1.2:0.3, hereinafter denoted as q-OPV. The maximum PCE of $8.42 \pm 0.12\%$ was greater than those of other BHJ–OPVs including the $7.59 \pm 0.19\%$ for the reference binary OPV made up of PTB7:$PC_{71}BM$ (hereinafter denoted as b-OPV) and even the $8.20 \pm 0.08\%$ for the optimized ternary OPV made up of PTB7:PCDTBT:$PC_{71}BM$ (hereinafter denoted as t-OPV), in which the mean values were obtained from more than 16 cells for each type of cell and error ranges correspond to respective s.d. values. The optimized quaternary composition was found to be responsible for the maximum $J_{SC}$ and FF (refer to short-circuit current density-to-voltage ($J$–$V$) characteristics in Supplementary Fig. 7 and

Supplementary Table 2). It is noted that there exists a room for further performance enhancement via more elaborated tuning of the polymer-to-fullerene ratio (for example, PTB7:PCDTBT: PC71BM:PC61BM = 0.9:0.1:1.6:0.4) (Supplementary Fig. 8, Supplementary Table 3)[5].

It is also notable that the q-OPV exhibited enhanced external quantum efficiency (EQE) spectra compared with those of the t- and b-OPVs (Fig. 2e). The enhancement corresponded to the increase in the photocurrent generated in the q-OPV under the broad spectrum of solar irradiation. The photocurrent density-to-effective voltage ($J_{ph}$–$V_{eff}$) characteristics, shown in Fig. 2f, were in accordance with the EQE result. We calculated the maximum exciton generation rate ($G_{max}$) and dissociation probability ($P(E,T)$), where $E$ and $T$ denote the electric field and temperature, respectively, to quantify the photocurrent generation and its efficiency from the $J_{ph}$–$V_{eff}$ data (Supplementary Table 4)[28,29]. The calculated $G_{max}$ agreed with the optically simulated value based on the E-field distribution in the active layer (Supplementary Figs 3c,d). Moreover, the quaternary blend exhibited a nearly 100% dissociation probability (97.36%). The improved dissociation property in the quaternary device was primarily associated with the decreased domain size (and hence increased D–A interfacial areas) in the blend, which is discussed below in detail. The device characterizations revealed the advantages of the q-OPV for attaining superior photovoltaic performances.

**Suppressing the early loss in performance.** Having confirmed the sufficiently high efficiency of the q-OPV, we then examined the stability of the OPV under elevated temperatures. The performance decay in OPVs is commonly characterized by two stages: a drastic exponential decline in performance at an early stage of operation (known as burn-in loss period), followed by a gradual linear decay on a longer timescale[30]. As shown in Fig. 3, b-, t- and q-OPVs operating at 65 °C were observed to display the

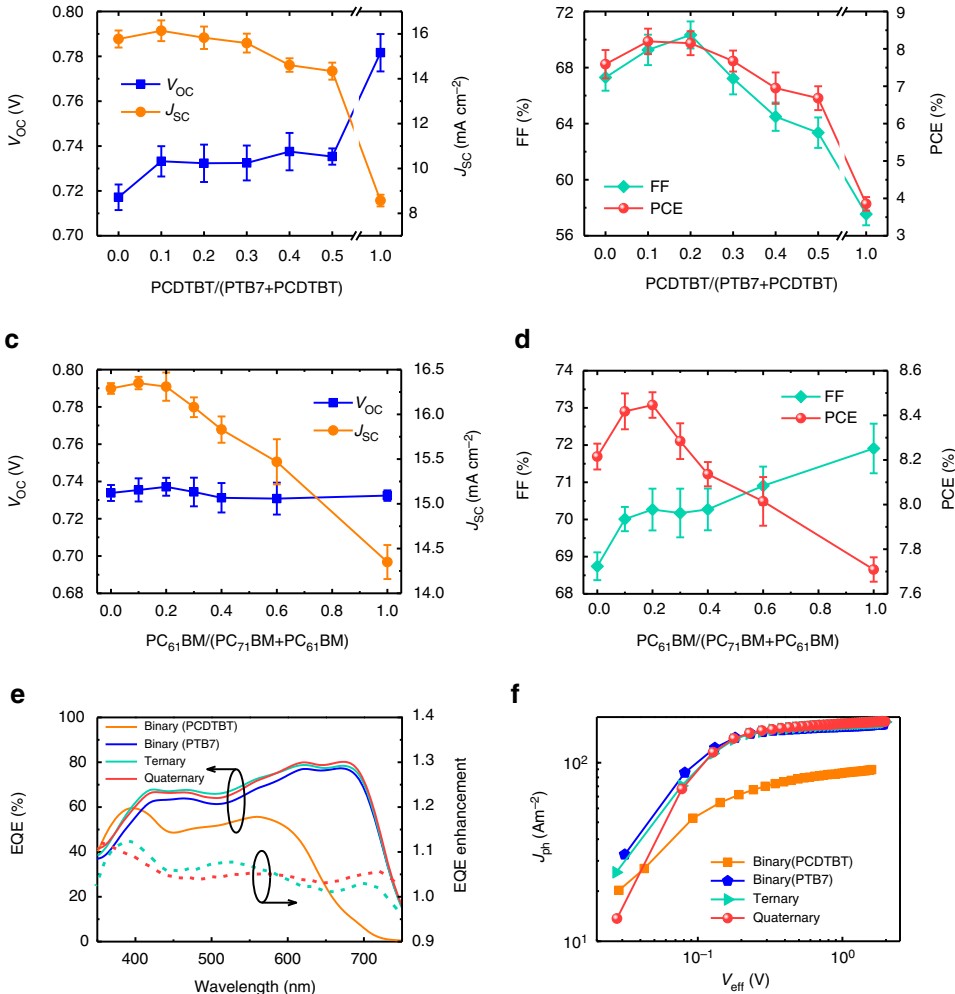

**Figure 2 | Optimization of compositions of the donors and acceptors in the q-OPV.** Photovoltaic parameters as a function of PCDTBT concentration ($x$, $0 \leq x \leq 1$) in PTB7:PCDTBT:PC$_{71}$BM ($1-x$:$x$:1.5) blends (**a**,**b**) and PC$_{61}$BM concentration ($y$, $0 \leq y \leq 1.5$) in PTB7:PCDTBT:PC$_{71}$BM:PC$_{61}$BM ($0.9$:$0.1$:$1.5 - y$: $y$) blends (**c**,**d**). The average and s.d. values in figures **a**–**d** were calculated using data from more than 16 cells. (**e**) EQE spectra and (**f**) $J_{ph}$–$V_{eff}$ characteristics of the b-, t- and q-OPVs. The EQE enhancement indicates the ratio of ternary (dotted turquoise line) or quaternary (dotted red line) devices to the PTB7-based binary device.

expected exponential decays in photovoltaic parameters at the early stages of exposure (the average and s.d. values obtained from more than 12 devices). The operating temperature is within the usual range of temperatures that outdoor applications of PVs are exposed to, and is comparable to (or higher than) the $T_g$ of typical conjugated polymers[12,31–34]. Details for the measurement are provided in 'Methods' section. The thermal stress at such 'real' OPV operating temperatures initially induces nanoscale grain growth rather than micron-size features[35]. As a result, we assumed that the nanoscale grain growth and structural disorder increase mainly contributed to the observed abrupt decrease in $J_{SC}$ and FF[18,36,37]. Compared with other parameters, $V_{OC}$ appeared to be nearly constant throughout the operation, which can be explained by the $V_{OC}$ being independent of the nano-grain growth[38]. Consequently, the q-OPV retained a PCE of up to 84.7% of the initial value even after 8 h of operation, and this value was much higher than the corresponding 72.1% and 65.6% values for the t- and b-OPVs, respectively. Interestingly, a thermal-dependent property showed that the performance decay of the b-OPV was accelerated at around 65 °C while the q-OPV displayed a better resistance to the decay even at elevated temperatures over 65 °C (Supplementary Fig. 9). The enhanced

internal quantum efficiency (IQE) spectrum in the q-OPV was consistent with the $J - V$ result regardless of the duration of the operation (see time-dependent IQE spectra and IQE–PCE characteristics in Supplementary Fig. 10). Considering that IQE is associated with the internal carrier transport process, which is strongly governed by nanomorphology, the q-OPV was apparently superior in suppressing the undesirable nanoscale crystallization and aggregation of the donor polymers fullerenes, respectively, in the early stages of the photovoltaic operation.

Then, to verify that the nano-grain growth was impeded, we carried out a two-dimensional grazing-incidence wide-angle X-ray scattering (2D GIWAXS) analysis of the active layer measured at different operation times. As shown in Fig. 4a–f and Supplementary Fig. 11, we observed common scattering patterns at $q_z \approx 0.35\,\text{Å}^{-1}$, corresponding to the Bragg diffraction peak (out-of-plane scattering in (100) direction) of the crystallized PTB7 (ref. 3), and ring patterns at $q \approx 1.32\,\text{Å}^{-1}$, corresponding to the aggregation of fullerene derivatives: these 2D scattering patterns were assigned as a reference for further characterizations. A detailed characterization of the scattering peak positions and full-width at half-maximum (FWHM) values is provided in Supplementary Fig. 12, Supplementary Table 5. Nano-grain size

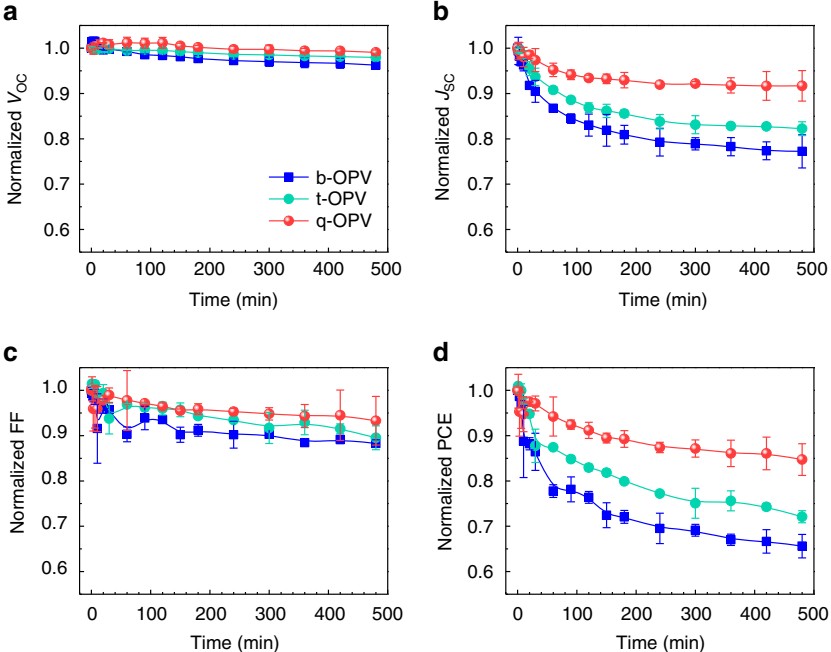

**Figure 3 | Suppressed initial photovoltaic performance loss in the q-OPV.** (**a**) $V_{OC}$, (**b**) $J_{SC}$, (**c**) FF and (**d**) PCE as a function of thermal treatment time (up to 8 h) at 65 °C for b-, t- and q-OPVs. The mean values with s.d. were obtained from more than 12 cells.

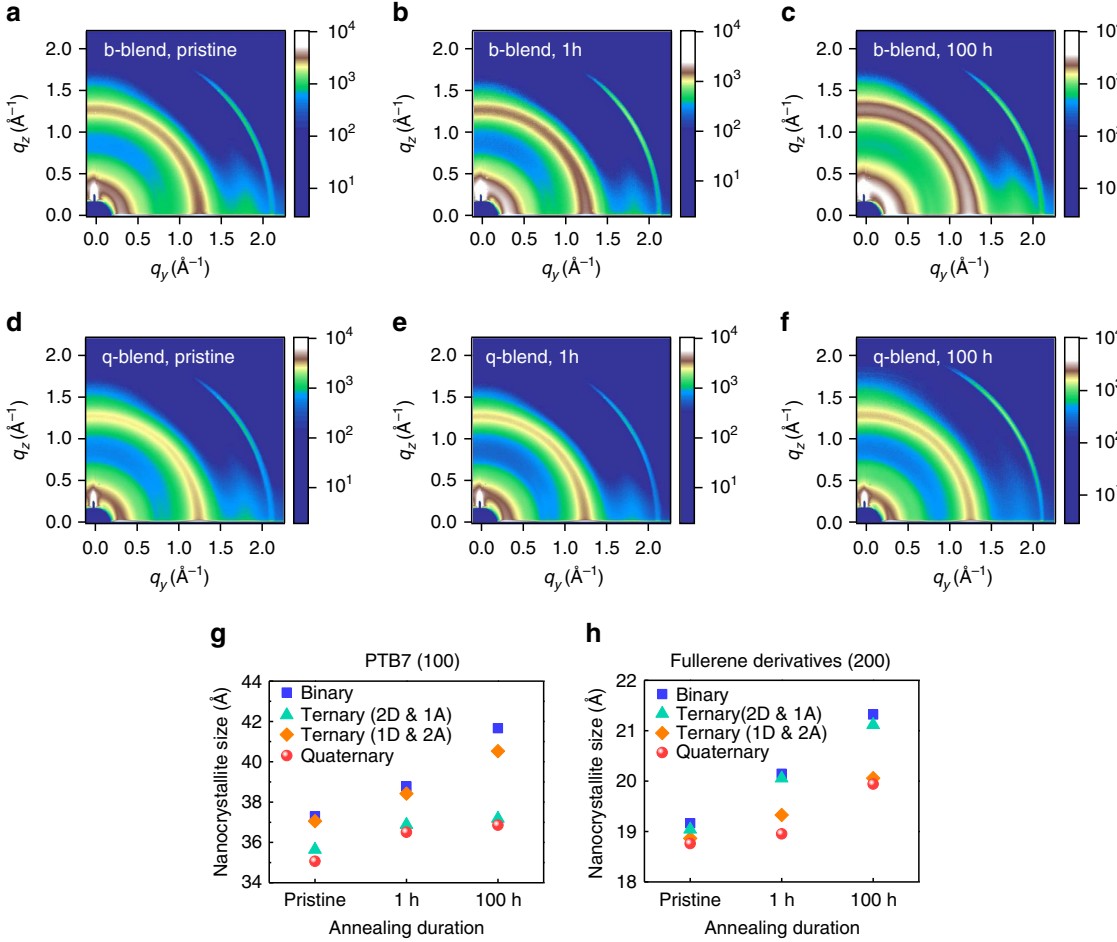

**Figure 4 | Nanoscale domain size analysis.** (**a–f**) 2D GIWAXS patterns of b- and q-blends for different annealing durations at 65 °C. Nanocrystallite size as a function of thermal treatment time for b-, t- (2D & 1A or 1D & 2A) and q-blends according to (**g**) PTB7 (100) and (**h**) fullerene derivatives (200). The active layer of the ternary device (2D & 1A) consisted of PTB7:PCDTBT:PC$_{71}$BM (0.9:0.1:1.5), while the ternary device (1D & 2A) was based on PTB7:PC$_{71}$BM:PC$_{61}$BM (1.0:1.2:0.3).

was calculated from the measured FWHM values by using the Scherrer equation[39,40]. As apparent from the calculated results, the nanoscale grain growth rate of both the polymers and fullerenes in the q-OPV was notably smaller than those of the t- and b-OPVs. Interestingly, the nanoscale crystallization of PTB7 was slowed down by the incorporation of PCDTBT (Fig. 4g), while the addition of $PC_{71}BM$ was less effective in impeding the PTB7 nanocrystallization. Similarly, the nanoscale aggregation size of fullerene derivative ($PC_{71}BM$) was also

notably reduced when another fullerene derivative ($PC_{61}BM$) was included (Fig. 4h), while the addition of PCDTBT was less effective in slowing down the fullerene nanocrystallization. Both the nanoscale grain and its growth rate were effectively reduced in the q-OPV compared with the t- and b-OPVs. Also, note that the increases in both hole ($\mu_h$) and electron mobilities ($\mu_e$) were suppressed for the quaternary device, but not for the binary and ternary devices (Supplementary Figs 6b–d). Considering that the mobilities increased proportionally to the crystal domain size of

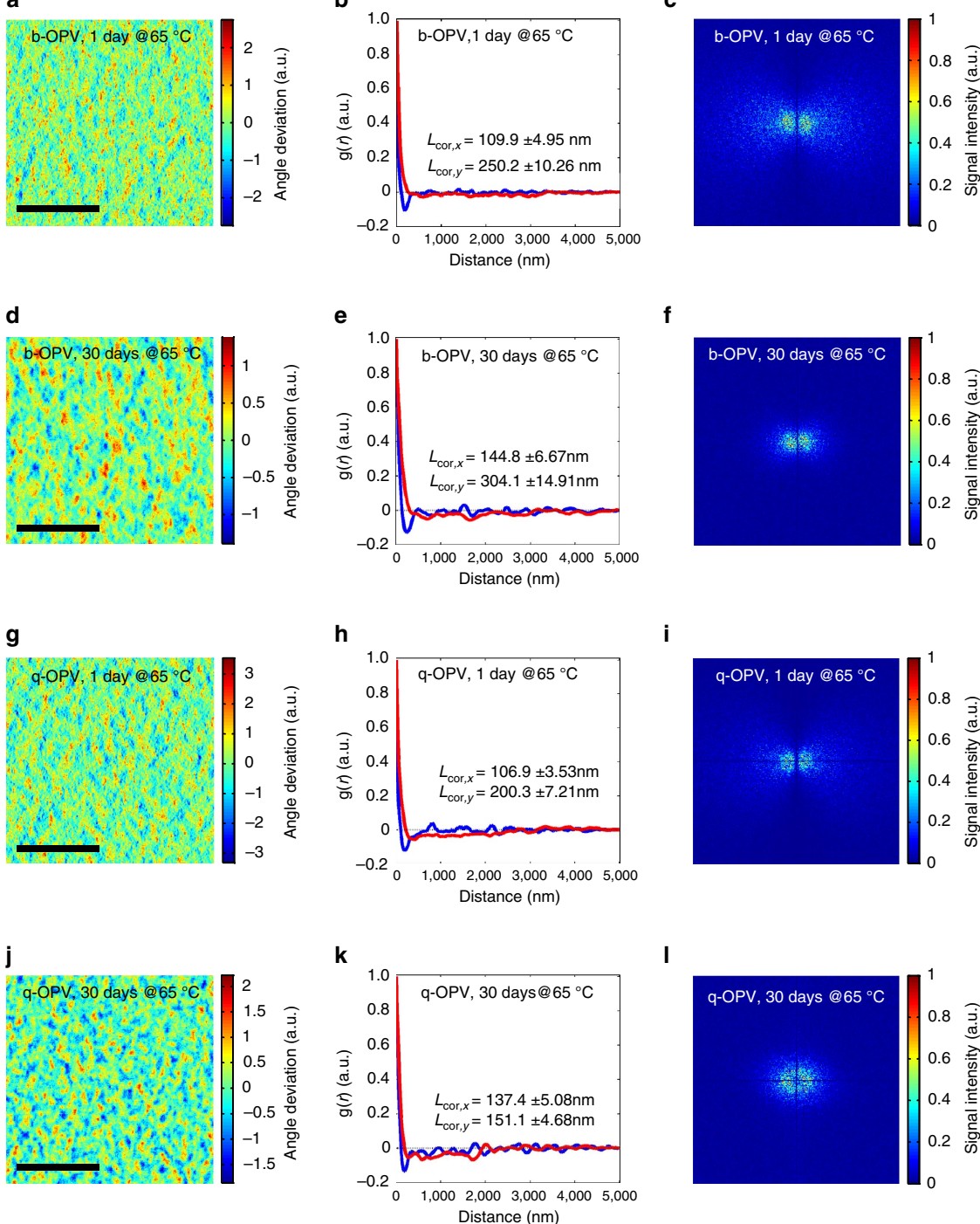

**Figure 5 | Long-term morphological stability analysis under thermal treatment.** A set of 2D AFM phase images, pair-correlation functions ($g(r)$), and 2D FFT profiles for the b- and q-OPVs with varying annealing durations at 65 °C: (**a–c**) b-OPV for 1 day, (**d–f**) b-OPV for 30 days, (**g–i**) q-OPV for 1 day and (**j–l**) q-OPV for 30 days. For the AFM data, colour bars denote the normalized orientational angle deviation and the scan area was $5 \times 5 \, \mu m^2$ (scale bars, 2 μm). The error ranges corresponding to the s.d. in **b,e,h,k** were obtained from AFM analysis of five samples.

donors and acceptors[41], this result clearly indicated a substantial inhibition of nanocrystallite growth in the quaternary device. Moreover, the quaternary device exhibited relatively balanced mobilities during the one day thermal treatment, also indicative of balanced charge transport and ideal domain size (Supplementary Fig. 6b)[27].

To uncover the origin of the reduction in the nanoscale grain growth rate in the q-OPVs, we used the Raman spectroscopy technique, which can explore vibrational modes of molecules and provide insight into polymer ordering and domain segregation[42,43] (details for the experimental measurements and analytical calculations are provided in Supplementary Methods). As shown in Supplementary Figs 13a–d, we assigned the vibrational normal mode at 1,441 cm$^{-1}$ to the vibrations of for hydrogen and carbon atoms of the conjugated rings of PTB7 and that at 1,457 cm$^{-1}$ to the vibrations of hydrogen and carbon atoms of the side-chain of $PC_{71}BM$ based on the first-principle calculations of the single molecule (Supplementary Figs 13e,f). A decrease of the Raman peaks with time was observed, indicative of a packing rearrangement of the molecules leading to the suppression of the vibration mode. Such a rearrangement is strongly related to the mechanism of phase segregation in the b-OPV upon thermal annealing[44]. In contrast, the peak intensities of the two vibrational modes were distinctively maintained in the q-OPV. Combined with the 2D GIWAXS data, the Raman spectroscopy data implied that the introduction of PCDTBT and $PC_{61}BM$ to a certain extent effectively inhibited the growth of the D and A components.

**Long-term stability of the q-OPV.** Following the drastic reduction in the performance of the devices at first use (for example, within about 1 day), their performances then exhibited a moderate decrease over a longer timescale of more than 30 days. We had conjectured that the long-term performance decay mechanism (after 1 day) would differ from that within one day. We, therefore, assigned the OPVs annealed for one day as a reference for long-term stability analysis. In contrast to the mechanism for the initial (about 1 day) decay, the main mechanism for the long-term decay in the photovoltaic performances would be the diffusion-limited phase separation driven by coalescence, which is a process governed by spinodal decomposition of the immiscible blends[45]. To compare in detail the phase-separated morphologies in the BHJs, we extracted the correlation length scale from the blends, which is strongly associated with the overall domain size (as discussed below). Figure 5a,d,g,j shows images of the 2D distributions of the orientation angle of the phase ($\varphi$), which were experimentally determined using atomic force microscopy (AFM) (see raw AFM

images in Supplementary Fig. 14). By applying the 2D phase value ($\varphi(r)$) to the pair-correlation function ($g(r)$), we obtained the correlation lengths in the $x$ ($L_{cor,x}$) and $y$ ($L_{cor,y}$) directions (Fig. 5b,e,h,k). A considerably narrow s.d. range of 3.21–4.89% was obtained from five samples, indicating the statistical significance of the measurement with internally consistent result (Supplementary Table 6). Details for the calculation of the correlation lengths can be found in 'Methods' section. The correlation length was found to increase in both the $x$ and $y$ directions more prominently in the b-OPV than in the q-OPV. This result was supported by analysing the 2D fast Fourier transform (2D FFT) patterns of the AFM phase image. As shown in Fig. 5c,f,i,l, the radii of the concentric rings in the 2D FFT pattern of the q-OPV were maintained during thermal treatment at a larger value than were those in the b-OPV-derived 2D FFT pattern, implying suppressed phase separation in the q-OPV.

As a consequence of those analyses, the q-OPV was found to be advantageous in retaining a high PCE for the extremely extendable operation duration (Fig. 6a). As the figure indicates, the q-OPV exhibited a strong resistance to the performance reduction even after a one-month operation at 65 °C (for example, >95% of the reference PCE). In contrast, the PCE of b-OPV fell to <80% of the reference PCE after this time period (time-dependent $V_{OC}$, $J_{SC}$ and FF of b- and q-OPVs in Supplementary Fig. 15). Interestingly, the notable inhibition of phase separation in the q-OPV also occurred in harsh temperature conditions. The q-OPV after one month at 120 °C retained 72.4% of its reference PCE, in contrast to a retention of only about 58.3% for the b-OPV (Fig. 6b). The 120 °C temperature sufficiently exceeds the operating temperature of photovoltaics running outside, which can be as high as 95 °C (ref. 12). Details for the correlation length scale as a function of the operating temperature can be found in Supplementary Fig. 16.

Next, from the calculated correlation length scale, we obtained the overall average domain size, $H_{inter}$, according to the equation $H_{inter} = (L_{cor,x}^{-2} + L_{cor,y}^{-2})^{-1/2}/2\pi$. Supplementary Table 6 compares $L_{cor,x}$, $L_{cor,y}$ and $H_{inter}$ of the b- and q-BHJs measured by AFM imaging of five different samples each. As shown in Fig. 7a, the q-OPV exhibited a notably smaller average phase-separated domain as well as suppressed domain growth after 30 days compared with the b-OPV (for example, an increase in $H_{inter}$ by 7.79% for the q-OPV versus 30.04% for the b-OPV at 65 °C). The q-OPV also exhibited such beneficial features under the extremely harsh temperature condition of 120 °C (Supplementary Fig. 17). For the multi-component mixture, for example, that having more than one D–A pair, it is known that the difference between the diffusivities of each component induces a kinetic trapping effect that slows down the separation rate[45]. For the q-OPV, the slowed-down phase separation also gave rise to a wider size

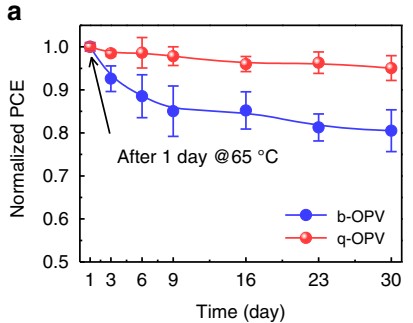
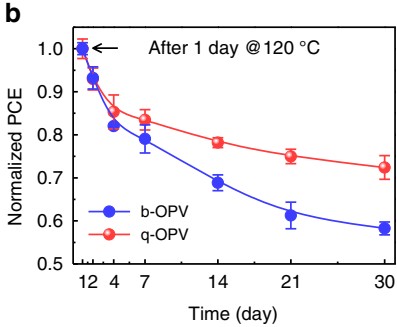

**Figure 6 | Time-dependent PCE decay after burn-in loss period under thermal conditions.** Time-dependent PCE of b- and q-OPVs relative to the reference PCE under the (**a**) moderate annealing temperature of 65 °C and (**b**) harsh annealing temperature of 120 °C for 30 days. The average values and error bars corresponding to the s.d. were obtained from more than 12 cells.

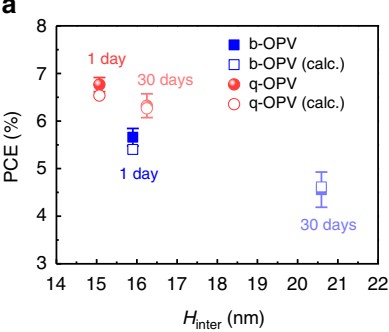
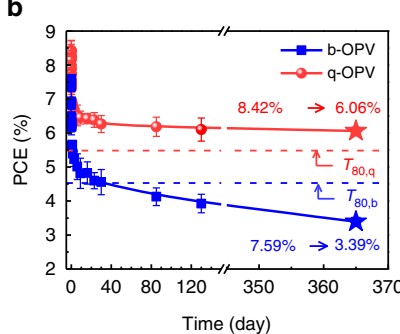

**Figure 7 | Long-term durability.** (**a**) Time-dependent PCE–$H_{inter}$ characteristics of b- and q-OPVs. The calculated PCE values were derived from the time-dependent $H_{inter}$ by using the modified drift-diffusion equation together with the Poisson equation. (**b**) Long-term PCE decay of b- and q-OPVs at 65 °C. The PCEs after one year (marked with star) are theoretically predicted values based on the PCE decay as a function of $H_{inter}$ kinetics. The horizontal dashed lines indicate the $T_{80}$ point (defined as the point at which the PCE has fallen to 80% of the value obtained after one day operation)[30] for the b- and q-OPVs. The combined experimental and calculated results suggested that the q-OPV after one year would display a PCE of 6.06%, retaining 72.0% of its initial value, whereas the b-OPV would suffer from a much more substantial PCE loss, retaining 44.7% of its initial value.

distribution of nanodomains than did the b-OPV, which was also attributed to the greater difference between the diffusivities of D and A. This greater difference resulted in the kinetic trapping effect, which decreased the rate of phase separation. To compare the rates of phase separation of the b- and q-OPVs, we employed the well-known kinetic model for the phase-separated domain governed by spinodal decomposition such that $H_{inter}(t) = H_{inter}(t_0) + C(t - t_0)^{1/3}$, where $C$ is the kinetic factor related to the Onsager mobility of the material and $t_0$ is the reference time (chose here to be one day)[45]. We found a $C$ of 0.382 day$^{-1/3}$ for the q-OPV, a value 4 times smaller than the 1.52 day$^{-1/3}$ for the b-OPV. Notably, the phase separation rate, $\Gamma_c$, can be derived from the relationship $\Gamma_c = 1/t_c$, where the characteristic timescale for the phase separation, $t_c$, can be obtained from the relationship $t_c = t_0 + C^{-3}$ based on $H_{inter}(t) = H_{inter}(t_0) + [(t - t_0)/(t_c - t_0)]^{1/3}$. Quantitatively, using the calculated $C$, we found that the phase separation rate for the q-OPV was much slower than that for the b-OPV by a factor of 14.7 (for example, $\Gamma_c$ of 0.053 day$^{-1}$ for the q-OPV versus 0.779 day$^{-1}$ for the b-OPV). It has been well known that the domain size and its growth rate are strongly associated with the photovoltaic performance[8–11]. We therefore further proceeded to numerically obtain the photovoltaic parameters that depended on the $H_{inter}$, which influences the exciton transport dynamics. To this end, we employed the modified drift-diffusion equation for the charge carriers and the Poisson equation for the electric potential, which used the exciton generation rate as previously provided in Supplementary Note 2 and Supplementary Fig. 3. From the numerical calculations, the time-dependent $V_{OC}$, $J_{SC}$, FF and PCE values can be obtained from the $H_{inter}$ values (details for the procedure and results can be found in Supplementary Note 3 and Supplementary Fig. 18). Interestingly, as Fig. 7a indicates, these calculated PCEs well matched the experimentally measured values.

As discussed above, the employed model could be statistically verified in analysing the effect of the change in morphology on the photovoltaic parameters. We further applied the model to anticipate the long-term (for example, one year) thermal stability of the q-OPV. By extrapolating the $H_{inter}$, we can obtain the photovoltaic parameters as well as the life expectancy of OPVs for such a long duration operation. Figure 7b shows the experimentally measured PCE (during more than 130 days) and the calculated PCE (after one year). The q-OPV was expected to retain more than 72.0% of its initial PCE after one year of operation at 65 °C and hence display a PCE of 6.06%. In contrast,

at the same conditions, the b-OPV was expected to retain only 44.7% of its initial PCE and hence display a PCE of only 3.39%. Notably, it is strongly expected that the q-OPV will not reach the $T_{80,q}$ point (defined as the point at which the PCE has fallen to 80% of the value obtained after one day operation)[30] even after one year of operation, whereas the b-OPV was estimated to reach the $T_{80,b}$ point within one month. Indeed, we found the $T_{80,q}$ of 4.20–4.36 years based on either a simple linear extrapolation or the theoretical model-based nonlinear extrapolation. Therefore, a highly extended life expectancy of more than several years can be anticipated for the q-OPV. For real outdoor application approach, long-term stability of the devices was further explored under light exposure (1 Sun, AM 1.5 condition). As provided in Supplementary Fig. 19, our q-OPV exhibited significantly better performance than that of the b-OPV under the photo-induced degradation conditions (5.15 versus 1.88% in the PCE after 21 days). Also, it is noteworthy that our q-OPV exhibited higher PCE throughout the duration of the thermal treatment or light illumination compared with literature values (Supplementary Table 7). The long-term stability of the q-OPV and its superior photovoltaic performances strongly suggest that this OPV can be used in outdoor applications with commercially acceptable quality.

## Discussion

In summary, we developed a novel q-BHJ, which consists of two donors and two acceptors, for photovoltaic applications and that display both high efficiency and long-term stability. The q-BHJ-based OPV (q-OPV) with an optimized composition exhibited a higher PCE, at 8.42%, than did the binary or ternary counterparts thanks to its broadened spectral response and improved internal photon-to-current conversion process. We also found the extended lifetime of the q-OPV to be due mainly to its enhanced long-term stability against thermal heating, which in turn we attributed to sufficient suppression of the growth of the domain size and resultant phase separation between the components. A satisfactorily sustained PCE of up to 6.10% was obtained after 130 days of operation with theoretically expected PCE of up to 6.06% after one year of operation at 65 °C. The q-OPV also exhibited a dramatic resistance to the loss of PCE at the extremely harsh operation temperature of 120 °C. The strongly enhanced long-term and thermal stability of the present q-OPV in conjunction with its high efficiency should substantially promote the applicability of this OPV for outdoor purposes.

## Methods

**OPV fabrication.** PCDTBT (molecular weight (Mw) of 57,000, 1-material), PTB7 (Mw of 115,000, 1-material), $PC_{71}BM$ (>99%, Nano-C) and $PC_{61}BM$ (>99.5%, Nano-C) were used as received. The device fabrication was based on the protocol optimized for the PTB7-based binary OPV. First, each of various ($x$ mg, $0 < x < 10$ mg) amounts of PCDTBT was dissolved in 970 μl of chlorobenzene (anhydrous, 99.8%, Sigma-Aldrich) and magnetically stirred at 80 °C for 72 h. PTB7 (10−$x$ mg), $PC_{61}BM$ ($y$ mg, $0 < y < 15$ mg), $PC_{71}BM$ (15 − $y$ mg) and 30 μl of 1,8-diodooctane (DIO, 98%, Sigma-Aldrich) were mixed with the fully blended PCDTBT solution, followed by magnetic stirring at 50 °C for 24 h. A solution of 0.5 mg ml$^{-1}$ PFN (1-material) and 2 μl ml$^{-1}$ acetic acid in methanol was spin-coated on ITO-coated glass and dried for 1 h at room temperature. The b-, t-, or q-BHJ blend was spin-coated on top of the PFN layer to form an active layer with a globally fixed thickness of ca. 90 ± 5 nm, in which the mean value was obtained from more than four scanning electron microscopy images for each sample and the error range corresponds to the s.d. The solvent was dried in a vacuum chamber overnight before top electrode deposition. An 8-nm-thick layer of $MoO_3$ and 100-nm-thick layer of Ag were then thermally evaporated through a shadow mask. The active area of the devices (0.116 cm$^2$) was defined by the overlap between the top and bottom electrodes. The complete OPVs were sealed in a nitrogen-filled glove box using an encapsulation glass immediately after the top electrode deposition.

**OPV characterization and long-term stability test.** The $J$–$V$ characteristics of the OPVs were recorded using a source meter (Keithley 2400) under an Air Mass 1.5 Global (AM 1.5G) illumination with an intensity of 100 mW cm$^{-2}$ (forward scan direction). White light was provided by a solar simulator (XES-301S, SAN-EI ELECTRIC) with a 300 W Xe lamp. The solar spectrum was modulated using an AM 1.5G filter without an additional filter. The light source was carefully adjusted by using a silicon reference cell calibrated by the National Renewable Energy Laboratory (NREL). We considered mean values of 3 respective measurements to minimize experimental errors. The main aim of this study is to investigate the enhanced long-term morphological stability in the q-OPV. Therefore, we did not consider a spectral mismatch factor, and the mask/aperture was not used for measurements. Instead, we attempted to secure fair comparison of long-term performance between b- and q-OPVs at the identical conditions. The EQE spectra were monitored using a K3100 Spectral IPCE Measurement System equipped with a 300 W Xe light source and monochromator (McScience). The $J_{SC}$ can be theoretically calculated by integrating the product of the incident photon flux density $F(\lambda)$ and $EQE(\lambda)$ of the device over the wavelength ($\lambda$) of the incident light, $J_{SC} = \int qF(\lambda)EQE(\lambda)d\lambda$, where $q$ is the elementary charge. We confirmed that the experimentally measured $J_{SC}$ was highly comparable to the EQE-derived $J_{SC}$. The IQE spectra were obtained based on IQE = EQE/(1 − $R$ − parasitic absorption), where the experimentally obtained spectral EQE and $R$ values were used (refer to parasitic absorption spectra in Supplementary Fig. 3b as well)[46]. The film absorption spectra of the active layers prepared on a quartz substrate were obtained using a Cary 5,000 ultraviolet–visible–NIR spectrophotometer (Varian). For the thermal stability test, the devices with full encapsulation were stored on a digital hot plate preheated to a desired temperature (65 or 120 °C) in dark ambient conditions, and the photovoltaic performance was obtained repeatedly under AM 1.5G illumination (repeated cycles of dark thermal annealing and device testing at room temperature). We assumed that the performance decay primarily resulted from the degradation within the active layer under the thermal stress because effects from other external factors (for example, exposure to oxygen or humidity) were minimized after encapsulation. All measurements were conducted using encapsulated devices under ambient conditions unless otherwise stated (controlled temperature of 25 °C and relative humidity of 50%).

**2D GIWAXS analysis.** GIWAXS measurements were taken at the Pohang Accelerator Laboratory using the 5A beam line and the samples for measurements were prepared as follows. The polymer-fullerene blends with different compositions of materials were coated on ITO-glass/PFN substrate, followed by encapsulation under a nitrogen atmosphere before annealing. The angle of incident X-rays was 0.13° and the incident photon energy was 11.57 keV. To minimize air scattering, samples were mounted in a helium ambient chamber. The in-plane GIWAXS profiles were fitted to a superposition of four Pearson VII functions for organic materials peaks and one exponentially decaying profile for the background. The peaks were assigned based on the previous study[39]. The fitting results are shown in Supplementary Fig. 12 and Supplementary Table 5. The nanocrystallite size was calculated by applying the Scherrer equation using FWHM values and a shape factor of 0.94.

**Numerical domain size calculation.** For the detailed analysis of the phase-separated morphology of the BHJ active layer, we extracted the domain size by image-analysis of the 2D distribution of the orientation angle of the phase ($\varphi$) measured by AFM. The tapping-mode phase images of the active layers were obtained using an XE-100 AFM (Park Systems). Considering that the difference in the phase angle is proportional to the compositional difference, it is instructive to quantify the spatial scale of the composition distribution using the pair-correlation function, $g(r)$. This function, defined by

$$g(r) = \frac{1}{N}\sum_{r'}\langle \bar{\varphi}(r+r')\bar{\varphi}(r')\rangle \tag{1}$$

allows for a determination of the long-range order of the system. In equation (1) $r$ is the 2D coordinates in the system, the bracket denotes values averaged over the system, and $\bar{\varphi} = \varphi - \varphi_{avg}$, where $\varphi_{avg}$ denotes the average value of $\varphi$. The obtained quantitative measure for the long-range order corresponds to the average domain size in the $i$th direction ($i = x$ or $y$), $L_{cor,i}$, which can be obtained by calculating the smallest value of $r$ satisfying $g(x) = 0$ and $g(y) = 0$, respectively. Then, the overall average domain size, $H_{inter}$, can be obtained using the reciprocal relationship $H_{inter} = (L_{cor,x}^{-2} + L_{cor,y}^{-2})^{-1/2}/2\pi$.

**Data availability.** The authors declare that the data supporting the findings of this study are available within the paper and its Supplementary Information files.

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

## Acknowledgements

This research was supported by Basic Science Research Program and Pioneer Research Center Program through the National Research Foundation of Korea, which is funded by the Ministry of Science, ICT, and Future Planning (NRF-2016R1C1B2014644, and NRF-2016M3C1A3909138). This work was also partly supported by the Energy Technology Development Program of the Korea Institute of Energy Technology Evaluation and Planning (KETEP) grant (Nos 20143030011530 and 2015030012870) funded by the Korean government, and by KIST institutional research program (2E26410).

## Author contributions

M.N. and M.C. conceived the main idea, designed and performed the experiments, and wrote the manuscript. H.H.L. conducted the X-ray diffraction measurement and analysis. K.H. performed the Raman analysis and first-principle calculations. K.-T.L. and J.Y. carried out the optical performance characterization. I.K.H. participated in the data analysis and discussion. S.J.K. conducted the optical simulations and kinetic modelling. D.-H.K. supervised the project. All authors discussed the results and commented on the manuscript.

## Additional information

**Competing financial interests:** The authors declare no competing financial interests.

