## [Peer Review File · Nature Communications]

Reviewers' comments:

Reviewer #1 (Remarks to the Author):

Overall, the authors have done an elaborate and comprehensive study on a novel quaternary blend system, which demonstrated an improved device performance and enhanced thermal stability at elevated temperatures.

The following are our specific comments.

1. Line 101-112: The effort to optimize the quaternary blend by first varying the PTB7:PCDTBT:PC71BM ratio highly resembles the approach published by Gupta et al. (*Adv. Mater.* 2015, 27, 4398). Gupta et al. further raised their PC71BM content such that their ternary PTB7:PCDTBT:PC71BM = 0.7:0.3:2 gave a higher PCE of 8.9%, while their binary PTB7:PC71BM = 1:1.5 only gave PCE of 6.8%. Have the authors considered to further optimizing the fullerene content as well?

2. Line 122-125: The authors noted that the coexistence of PC61BM and PC71BM in the quaternary device might result in a more balanced hole-to-electron mobility ratio (supported by their charge-carrier mobility measurement). Is this increase in electron mobility in the quaternary device caused by the different (possibly higher) electron mobility of PC61BM or a more interconnected between both phases of fullerene derivatives? Are there ways to verify this?

3. Line 221-225: The authors derived the correlation lengths from AFM images. Can the authors comment on the statistics of this calculation? We understand that AFM measurement can be quite localized (scan size of $5 \times 5 \mu\text{m}^2$). We are also aware that domain sizes of donor and acceptor phases are strongly associated with their respective correlation lengths from grazing-incidence X-ray measurement (Mukherjee et al., *Adv. Energy Mater.* 2015, 5, 1500877). The derivation of domain sizes from GIWAXS data might be able to give better statistics due to the larger exposed area. What are the authors' thoughts about this (since the manuscript also contains GIWAXS data)?

4. Line 217-218: To simulate the real thermal stability testing, we believe the devices ought to be soaked in light for a certain period, instead of heating the encapsulated device in dark ambient conditions; similar conditions are mentioned in Supplementary Table 5. Photo-induced degradation could also occur, and might be even more interesting for real-life applications. Have the authors attempted to compare the evolution of device parameters under constant illumination (for a fixed period) vs intermittent measurement of heated device that was kept in dark?

Reviewer #2 (Remarks to the Author):

In this work a Long-term efficient organic solar cell based on PTB7:PCDTBT:PC60BM:PC70BM blend system is reported. The authors have proved once again the multi-functionality of the multi-component systems such as ternary or quaternary blends as compared to their corresponding binary host systems. This important, as an elegant alternative approach for boosting the power conversion efficiency and enlarging the lifespan of solar cells, will have a significant effect on the commercialization of organic photovoltaic technology. Although impressive, still some open issues has to be addressed in this manuscript to reach the high standards of Nature communications. I can accept it just after a major revision.

- Introduction and stability sections: Why the author are insisting to report the performance within the first year? It is very common in the community to report the T80.

- Section "Device Structure and Optical Properties of Materials": How do the author claim a hole rely process under the cascade-energy-level formation? There is no study which proves that except a PL study which reveals the energy transfer from PCDTBT to PTB7. Even the improved IQE can origin from higher charge generation and improved transport. Since the morphology is modified and also mobility, for sure transport has a dominant role. To claim a cascade charge transfer between the polymers, they should prove it directly by employing for example photo-induced absorption spectroscopy.

- Section "Device Structure and Optical Properties of Materials": What is the motivation of the optical simulation with T-matrix method? Which information are obtained there which is not present in the usual UV-Vis spectroscopy results? I see it a useless piece of info and results in the manuscript which can be simply removed, except they state their motivation and additional achieved information more clearly. The same argument with Gmax compared to the results from simulation.

- Figure 6 and related discussion: This part is strange to me. The presented data in the first month comes from the experimental results and the rest from the theoretical calculation based on extrapolated Hintereisner? Why they don't extrapolate just the PCE achieved from experiments and estimate the T80? In figure 6a and SI 17d, there is a slight slope difference between the curves achieved from experiments and calculations. What is the difference of PCE at T80 for these two cases? I guess that this small slope difference will get much more pronounce after long-term calculations based on extrapolation.

- SI figure 16: Despite the more stable trend calculated for Hintereisner, PCE shows a faster drop at high temperature for the quaternary system as compared to the binary reference. It is not mentioned and addressed properly in the manuscript.

- According to literature, multi-component systems, ternaries or quaternaries, may show an

adjusted recombination mechanisms compared to their binary reference system. It will influence the lifetime of the solar cells, particularly under light. This important and relevant point should be shortly noticed in the introduction section and properly cited.

- Table 5 in SI: A comparison between the achieved results under dark condition to those results obtained under light, as well as encapsulated and not-encapsulated devices is really inaccurate and unfair comparison.

Reviewer #3 (Remarks to the Author):

Summary:

In recent years, ternary blend organic solar cells have been presented as a viable means of extending the light absorbance in organic solar cells. This study presents a quaternary blend solar cell with improved performance over the binary or ternary blend cells based on its components. This is due in large part to the extended light absorption and balanced carrier mobilities, as shown in UV-Vis, quantum efficiency measurements, and SCLC measurements. More importantly, the authors show that the quaternary blend cell has enhanced stability over the binary and ternary blends, maintaining 72% of the original performance after significant thermal aging. GIWAX studies, AFM, and simulations of domain growth indicate that this stability is due to reduced domain growth in the quaternary blend. Using multiple components to control the crystallization etc. is a novel idea.

This paper presents a thorough study of a quaternary blend organic solar cell. The improved performance over the binary and ternary blends is interesting in and of itself, but coupled with the enhanced stability, this represents an intriguing result. The reviewer recommends this paper for acceptance with the following minor revisions.

Minor Issues

- 1) When discussing the energy transfer between donor materials, the authors assert that the "combination of two donors was also beneficial for the energy transfer kinetics as revealed in photoluminescence." While, it is clear that energy or charge transfer is occurring between the two polymers, it is not clear that this aids the quenching as compared to the binary blends. To make this assertion, we would need to compare the PL of the binary blends. Additionally, the presented PL measurements are steady state, and do not explain anything about the kinetics of the systems.
- 2) In Figure 2.e, it is not clear what the "EQE Enhancement" is, or how it was calculated. Is this the enhancement compared to the binary blends? If so, which one?
- 3) Ln 196, The authors claim that the q-OPV shows improved carrier transport after thermal treatment. In reality, the q-OPV shows less absolute improvement in the carrier transport than the b-OPV or t-OPV, but this is indicative of balanced charge transport and ideal domain size. This

should be clarified.

4) While the paper is certainly readable, there are several phrases which are awkward or unclear, and should be edited grammatically. The manuscript would be more readable if it can go another round of professional proof-reading. A few examples:

- Ln 50: "In this standpoint" should be "From this standpoint", and the following sentence is unclear.
- Ln 93: "Accorded well" should be "agreed well" or something similar
- Ln 164-167: "temperature exhibited that the performance" is awkward, and the sentence is unclear in general
- Ln 235-236: "A very long time of use" and "Maintained quite well its performance"

Reviewers' comments:

Reviewer #1 (Remarks to the Author):

Overall, the authors have done an elaborate and comprehensive study on a novel quaternary blend system, which demonstrated an improved device performance and enhanced thermal stability at elevated temperatures.

Overall response: We authors appreciate valuable commentary and positive decision from the reviewer. We have concentrated on securing the stability results of the q-OPV for sufficiently long duration. Given the maximum period of time for the revision (3 months), we could extend the device stability test by measuring it at the point of 4 months operation (1 + 3 months, ~130 days). From the updated test, we confirmed that the long-term operation stability results agreed well again with our theoretical model (refer to modified Figure 6 b). Based on the updated results and the previous analysis, we could secure strongly conclusive evidence and more profound understanding on the stability enhancement of the proposed q-OPVs.

1. Line 101-112: The effort to optimize the quaternary blend by first varying the PTB7:PCDTBT:PC71BM ratio highly resembles the approach published by Gupta et al. (Adv. Mater. 2015, 27, 4398). Gupta et al. further raised their PC71BM content such that their ternary PTB7:PCDTBT:PC71BM = 0.7:0.3:2 gave a higher PCE of 8.9%, while their binary PTB7:PC71BM = 1:1.5 only gave PCE of 6.8%. Have the authors considered to further optimizing the fullerene content as well?

Response: We appreciate a thoughtful comment from the reviewer. As the reviewer suggested, we compared four types of OPVs with different D-A ratios. For the PTB7-based b-OPVs, the D-A ratio of 1:1.5 by weight produced a better PCE of 7.59% compared to 7.18% of the 1:2 case. In fact, the best binary D-A ratio of 1:1.5 by weight has been widely known for the PTB7-based BHJ OPVs in the community [Adv. Mater. 22, E135-E138, 2010 & Nature Photon. 6, 591-595, 2012]. Interestingly, we found that the q-OPV with a D-A weight ratio of 1:2 showed a PCE of 8.54% (V_{oc} of 0.74 V, J_{sc} of 16.46 mAcm⁻², and FF of 70.59%), which is slightly higher than 8.42% obtained from the 1:1.5 ratio q-OPV (V_{oc} of 0.74 V, J_{sc} of 16.31 mAcm⁻², and FF of 70.25%). This would explain the motivation taken by Gupta *et al.* who tried to raise the fullerene content in their ternary OPVs [Adv. Mater. 27,

4392-4404, 2015]. However, the performance of the 1:2 q-OPV could not exceed that of the 1:1.5 device. It is indeed very important that the polymer-to-fullerene ratio has a large influence on the morphological properties in the BHJ active layer. In this research, we aim to investigate morphological stability of the OPVs of the 1:1.5 ratio case that has been widely referenced. **For a reasonable comparison, we assigned the 1:1.5 q-OPV as a device for long-term stability analysis, which is relevant as a reference of the state-of-the-art PTB7-based BHJ OPVs** (Supplementary Table 7). According to the reviewer's suggestion, we have provided that the performance of the q-OPVs can be further improved by fine-tuning the ratios between components. For detailed information, we also additionally provided Fig. S8 and Table S3 in the revised Supplementary Information.

→ (Page 7) The optimized quaternary composition was found to be responsible for the maximum J_{sc} and FF (refer to short-circuit current density-to-voltage ($J-V$) characteristics in Supplementary Fig. 7 and Supplementary Table 1). **It is noted that there exists a room for further performance enhancement via more elaborated tuning of the polymer-to-fullerene ratio (e.g., PTB7:PCDTBT:PC71BM:PC61BM = 0.9:0.1:1.6:0.4) (refer to Supplementary Fig. 8 and Supplementary Table 3)**⁵.

Supplementary Fig. 8. D-A ratio-dependent PV properties. $J-V$ characteristics of (a) b-OPV and (b) q-OPV as a function of overall polymer(s)-to-fullerene(s) ratio.

Supplementary Table 3. Comparison of the photovoltaic parameters between the b- and q-OPVs with different overall D-A ratios of 1:1.5 or 1:2.

PTB7:PCDTBT:PC ₇₁ BM:PC ₆₁ BM	V_{oc} (V)	J_{sc} (mAcm ⁻²)	FF (%)	PCE (%)
1:0:1.5:0	0.72	15.77	67.29	7.59
	±0.01	±0.19	±0.94	±0.19
1:0:2.0:0	0.71	14.99	67.77	7.18
	±0.01	±0.12	±0.91	±0.23

0.9:0.1:1.2:0.3	0.74	16.31	70.25	8.42
	±0.01	±0.17	±0.56	±0.12
0.9:0.1:1.6:0.4	0.74	16.46	70.95	8.54
	±0.01	±0.13	±0.63	±0.15

2. Line 122-125: The authors noted that the coexistence of PC61BM and PC71BM in the quaternary device might result in a more balanced hole-to-electron mobility ratio (supported by their charge-carrier mobility measurement). Is this increase in electron mobility in the quaternary device caused by the different (possibly higher) electron mobility of PC61BM or a more interconnected between both phases of fullerene derivatives? Are there ways to verify this?

Response: As shown in Table S1 and Fig. S6 in the revised Supplementary Information, we experimentally found more balanced hole-to-electron mobility ratio as incorporating PC₆₁BM into the PTB7:PC₇₁BM blend. This is mainly attributed to the increased electron mobility. The increase in the electron mobility can result in the improvement in FF as previously reported by others [Adv. Mater., 2016, DOI: 10.1002/adma.201602067]. It has been further investigated that the PC₇₁BM-based (or PC₇₁BM-rich) BHJ devices show relatively lower FF values compared to that of the PC₆₁BM-based (or PC₆₁BM-rich) ones due to low mobility as well as relatively unbalanced mobilities. One of the factors underlying the more balanced mobilities by incorporating PC₆₁BM would lie in the better symmetric molecular structure of PC₆₁BM compared to PC₇₁BM [Adv. Energy Mater., 2014, 1401687]. Assuredly, we believe that there would be other factors (*e.g.*, morphological interconnection between fullerene derivatives), which can alter the mobility and consequently FF of the device. A detailed study employing advanced analysis tools (*e.g.*, X-ray scattering analysis or microscopic photonic analysis such as EELS/NSOM imaging) will be a subject of next independent research of ours in near future. We included the mobility data in Table S1 and Fig. S6 in the revised Supplementary Information, and the manuscript has been revised correspondingly as below.

→ (Page 6) Note, in particular, that the enhancement of the FF would be attributed to the balanced charge carrier mobilities or morphologically favorable alteration when the fraction of PC₆₁BM was increased (see hole-to-electron mobility ratio in Supplementary Fig. 6a and Supplementary Table 1)²⁷.

Supplementary Table 1. Comparison of carrier mobility between PTB7:PC₇₁BM binary and PTB7:PC₇₁BM:PC₆₁BM 2A ternary devices.

	μ_e [cm ² V ⁻¹ s ⁻¹]	μ_h [cm ² V ⁻¹ s ⁻¹]	μ_h / μ_e
PTB7:PC₇₁BM (1:1.5)	6.09×10^{-5}	1.99×10^{-4}	3.27
PTB7:PC₇₁BM:PC₆₁BM (1:0.8:0.2)	9.97×10^{-5}	2.02×10^{-4}	2.02

Supplementary Fig. 6. Charge carrier mobility. (a) Hole mobility (μ_h) to electron mobility (μ_e) ratios for pristine binary, ternary, and quaternary devices. The t(2D) device was composed of two donors and one acceptor (PTB7:PCDTBT:PC₇₁BM = 0.9:0.1:1.5), while the t(2A) device was composed of one donor and two acceptors (PTB7:PC₇₁BM:PC₆₁BM = 1.0:1.2:0.3). (b) The μ_h/μ_e ratio, (c) μ_h , and (d) μ_e at 65°C for various points up to 24 h.

3. Line 221-225: The authors derived the correlation lengths from AFM images. Can the authors comment on the statistics of this calculation? We understand that AFM measurement can be quite localized (scan size of 5 x 5 μm^2). We are also aware that domain sizes of donor and acceptor phases are strongly associated with their respective correlation lengths from grazing-incidence X-ray measurement (Mukherjee et al., Adv. Energy Mater. 2015, 5, 1500877). The derivation of domain

sizes from GIWAXS data might be able to give better statistics due to the larger exposed area. What are the authors' thoughts about this (since the manuscript also contains GIWAXS data)?

Response: As the reviewer commented, the typical dimension of the AFM image employed in our study was $5 \times 5 \mu\text{m}^2$, which would not be sufficient to represent the entire system of the BHJ. In order to secure the statistical reliability of the observed data, we measured the correlation lengths and the domain sizes of different samples at the identical conditions (*i.e.*, AFM examination of 5 samples of b- and q-BHJs at 65°C taken at 1 day and 30 days each). As provided in Supplementary Table 6 below, we found that the overall standard deviation range in the correlation lengths and the domain sizes is 3.21-4.89% depending on the system. **The error range was considerably narrow, indicating the statistical significance of the measurement with internally consistent result.** This is mainly because the domain size (*i.e.*, $H_{inter} \sim 15\text{-}21 \text{ nm}$ given 1 - 30 days of operation of the b- and q-BHJs) is considerably smaller than the image dimension ($5 \times 5 \mu\text{m}^2$). In a single image dimension, there are $\sim 10^5$ domains, which can be collectively represented by intensity distribution in the phase mode image taken by AFM, and consequently leads to the reduction in the error. We newly included the standard deviation of AFM images obtained from 5 different samples in Figure 5b, e, h, and k as well as in Supplementary Table 6.

[REDACTED]

→ (Page 10-11) By applying the 2D phase value ($\varphi(r)$) to the pair-correlation function ($g(r)$), we obtained the correlation lengths in the x ($L_{cor,x}$) and y ($L_{cor,y}$) directions (Figures 5b, e, h, and k). **A considerably narrow standard deviation range of 3.21 – 4.89% was obtained from 5 samples, indicating the statistical significance of the measurement with internally consistent result (Supplementary Table 6). Details for the calculation of the correlation lengths can be found in Methods.**

→ (Page 11) Next, from the calculated correlation length scale, we obtained the overall average

domain size, H_{inter} , according to the equation $H_{inter} = (L_{cor,x}^{-2} + L_{cor,y}^{-2})^{-1/2} / 2\pi$.

Supplementary Table 6 compares L_{cor} and H_{inter} of the b- and q-BHJs measured by AFM imaging of 5 different samples each.

Supplementary Table 6. Comparison of L_{cor} and H_{inter} of the b- and q-BHJs operated for 1 day and 30 days at 65°C measured by AFM images of 5 different samples each.

	Operation duration	$L_{cor,x}$ (nm)	$L_{cor,y}$ (nm)	H_{inter} (nm)
b-BHJ	1 day	109.9 ± 4.95	250.2 ± 10.26	16.01 ± 0.69
	30 days	144.8 ± 6.67	304.1 ± 14.91	20.82 ± 1.00
q-BHJ	1 day	106.9 ± 3.53	200.3 ± 7.21	15.01 ± 0.53
	30 days	137.4 ± 5.08	151.1 ± 4.68	16.18 ± 0.52

Figure 5. Long-term stability of b- and q-OPVs under thermal conditions. A set of 2D AFM phase images, pair-correlation functions ($g(r)$), and 2D FFT profiles for the b- and q-OPVs with varying annealing durations at the moderate temperature of 65°C: (a-c) b-OPV for 1 day, (d-f) b-OPV for 30 days, (g-i) q-OPV for 1 day, and (j-l) q-OPV for 30 days. For the AFM data, color bars denote the normalized phase angle and the scan area was $5 \times 5 \mu\text{m}^2$ (scale bars denote $2 \mu\text{m}$). **The standard deviation in (b), (e), (h), and (k) was obtained from AFM analysis of 5 samples.** Time-dependent PCE decay relative to the reference PCE under the (m) moderate annealing temperature of 65°C and (n) harsh annealing temperature of 120°C for 30 days. The mean values were obtained using data from more than 12 cells.

[REDACTED]

[REDACTED]

[REDACTED]

[REDACTED]	[REDACTED]	[REDACTED]	[REDACTED]	[REDACTED]	[REDACTED]
[REDACTED]	[REDACTED]	[REDACTED]	[REDACTED]	[REDACTED]	[REDACTED]
[REDACTED]	[REDACTED]	[REDACTED]	[REDACTED]	[REDACTED]	[REDACTED]
[REDACTED]	[REDACTED]	[REDACTED]	[REDACTED]	[REDACTED]	[REDACTED]

4. Line 217-218: To simulate the real thermal stability testing, we believe the devices ought to be soaked in light for a certain period, instead of heating the encapsulated device in dark ambient conditions; similar conditions are mentioned in Supplementary Table 5. Photo-induced degradation could also occur, and might be even more interesting for real-life applications. Have the authors attempted to compare the evolution of device parameters under constant illumination (for a fixed period) vs intermittent measurement of heated device that was kept in dark?

Response: We appreciate for the reviewer's suggestion. We carried out a long-term light soaking test under light exposure in ambient conditions (at 25°C and 50% relative humidity). The encapsulated devices were illuminated under AM 1.5G solar simulator with a 12 h light/dark illumination cycle for ~21 days. As shown in Fig. S19 in the revised Supplementary Information, the q-OPV retained more than 61.16% of its initial PCE after ~21 days of illumination (5.15% in the PCE), which was considerably higher than 24.76% maintenance in the b-OPV at the same conditions (1.88% in the PCE). **It is notable that our q-OPV exhibited a higher (or comparable) PCE in comparison with the previously reported state-of-the-art OPVs under the photo-induced degradation condition** (Table S7 in the revised Supplementary Information). The light soaking test results indicate the long-term thermal and photochemical stability of the q-OPV, which is advantageous to outdoor applications. As reviewer pointed out, solar illumination of the devices may influence PV stability due to photo-induced degradation of conductive polymers as well as morphological mutation of BHJ films. In this study, we aimed to deconvolute these effects by studying morphological stability in dark condition. Thus, to gain insight into the enhanced long-term morphological stability in q-BHJs, the main degradation factor was selectively limited to thermal stress, while other degradation parameters (*e.g.*, water, oxygen, light soaking, UV stress, *etc.*) were minimized by encapsulating the devices and then aging them in the dark. We newly included the light soaking result in Fig. S19 and Table S7 in the revised Supplementary Information. The modified parts in the manuscript are as below.

→ (Page 13) For real outdoor application approach, long-term stability of the devices was further explored under light exposure (1 Sun, AM 1.5 condition). As provided in Supplementary Fig. 19, our q-OPV exhibited significantly better performance than that of the b-OPV under the photo-induced degradation conditions (5.15% vs. 1.88% in the PCE after ~21 days). Also, it is noteworthy that our q-OPV exhibited higher PCE throughout the duration of the thermal treatment or light illumination compared to literature values (see Supplementary Table 7). The long-term stability of the q-OPV and its superior photovoltaic performances strongly suggest that this OPV can be used in outdoor applications with commercially acceptable quality.

Supplementary Fig. 19. Long-term photo-induced degradation test. Long-term PCE decay of b- and q-OPVs under AM 1.5G solar simulator with a 12 h light/dark illumination cycle for ~21 days.

Supplementary Table 7. Superior PCE sustainability of our q-OPV. Comparison of the PCE decay (the percentage of PCE decrease relative to the initial value) between our q-OPV and other state-of-the-art binary OPVs under diverse aging conditions (some PCE values estimated from the figure images, and not exactly stated in literature, are labeled with *ca.*).

D:A material	Device structure	Initial PCE	Last PCE	PCE loss	Degradation conditions (e.g., light, temperature, encapsulation)	Ref.
q-OPV	Inverted	8.42%	6.27% after 30 days (experimental) & 6.06% after one year (simulated)	25.53% & 28.03%	Dark, 65°C, with encapsulation	This study
b-OPV		7.59%	4.56% after 30 days (experimental) & 3.39% after one year (simulated)	39.92% & 55.34%		
q-OPV		8.42%	5.15% after ~21 days	38.84%		
b-OPV		7.59%	1.88% after ~21 days	75.23%		
P3HT:PC ₆₁ BM	Standard	3.0%	ca. 1.5% after 4700h	ca. 50%	Continuous illumination under a Sulphur plasma lamp, 50°C (testing chamber temperature), with encapsulation	18
P3HT:PC ₆₁ BM	Standard	3.7%	2.5% after 1000h	32.43%	Dark, 45°C, w/o encapsulation (inert measurement conditions)	19
P3HT:PC ₆₁ BM	Standard	3.2%	1.8% after 1000h	43.75%	Continuous illumination under a 150W Xenon lamp with AM 1.5G filter, 45°C, w/o encapsulation	19

(inert measurement conditions)						
P3HT:PC ₆₁ BM	Standard	4.0 ± 0.05%	ca. 2.92% after 4400 h (experimental) & ca. 2.72% after 3.1 years (simulated)	27% & 32%	Continuous illumination under a Sulfur plasma lamp (6000 K), 37°C, with encapsulation	20
PCDTBT:PC ₇₁ BM	Standard	5.5 ± 0.15%	ca. 3.74% after 4400h (experimental) & ca. 3.19% after 6.2 years (simulated)	32% & 42%		
PCDTBT:PC ₇₁ BM	Standard	7.04%	ca. 5.56% after 19500h (simulated) & ca. 5.63% after 650 days (experimental)	21.02% & 20.03%	Continuous illumination under a Sulphur plasma lamp (6000 K), room temperature, with encapsulation	21
PCDTBT:PC ₇₁ BM	Standard	6.50%	ca. 3.25% after 30 days	ca. 50%	Under ambient air conditions, w/o encapsulation	22
PCDTBT:PC ₇₁ BM	Standard	5.02%	3.54% after 4500h	29.48%	Continuous illumination under a halide lamp (1000 Wm ⁻²), 45°C, with encapsulation	23
PTB7:PC ₇₁ BM	Inverted	5.37%	ca. 3.33% after ca. 3500h	ca. 37.98%	Under ambient dark conditions, w/o encapsulation	24

Reviewer #2 (Remarks to the Author):

In this work a Long-term efficient organic solar cell based on PTB7:PCDTBT:PC60BM:PC70BM blend system is reported. The authors have proved once again the multi-functionality of the multi-component systems such as ternary or quaternary blends as compared to their corresponding binary host systems. This important, as an elegant alternative approach for boosting the power conversion efficiency and enlarging the lifespan of solar cells, will have a significant effect on the commercialization of organic photovoltaic technology. Although impressive, still some open issues has to be addressed in this manuscript to reach the high standards of Nature communications. I can accept it just after a major revision.

Overall response: We authors appreciate valuable commentary and positive decision from the reviewer. We have concentrated on securing the stability results of the q-OPV for sufficiently long duration. Given the maximum period of time for the revision (3 months), we could extend the device stability test by measuring it at the point of 4 months operation (1 + 3 months, ~130 days). From the updated test, we confirmed that the long-term operation stability results agreed well again with our theoretical model (refer to modified Figure 6 b). Based on the updated results and the previous analysis, we could secure strongly conclusive evidence and more profound understanding on the stability enhancement of the proposed q-OPVs.

1. Introduction and stability sections: Why the authors are insisting to report the performance within the first year? It is very common in the community to report the T₈₀.

Response: We appreciate the reviewer's sincere review and comments. We considered that depicted performance within the first year ensures more reliable result. Indeed, even T₈₀ for q-OPV was estimated as over one year while b-OPV was within a month. Thus we intended to conservatively evaluate morphological stability by comparing PV performance within one year. As reviewer suggested, we modified Figure 6b to clearly describe T₈₀ of devices which was defined as the point at which the PCE has fallen to 80% of the value obtained after one day operation [Adv. Energy Mater. 1, 491-494, 2011]. We found that the PCE of b-OPV decayed rapidly and reached its T₈₀ point only within one month. In contrast, we expect a highly extended life expectancy with more than 4 years of T₈₀ for the q-OPV based on the extrapolation method based on a theoretical model given in the manuscript. In addition, from reviewer's comment, we attempted to secure the stability data for the

period of time as long as possible. Thus we further examined both b- and q-OPVs under thermal annealing at 65°C for ~130 days (maximum period given for this revision). We found that the 130 days-operation results agreed well with the theoretically predicted curves (see modified Figure 6b). According to the reviewer's comment, we compared the PCE decay within the first year as well as T_{80} of b- and q-OPVs. The modified sentences and Figure 6b are as below.

→ (Page 13) Notably, it is strongly expected that the q-OPV will not reach the $T_{80,q}$ point (defined as the point at which the PCE has fallen to 80% of the value obtained after one day operation)³⁰ even after one year of operation, whereas the b-OPV was estimated to reach the $T_{80,b}$ point within one month. Indeed, we found that $T_{80,q}$ of 4.20 - 4.36 years based on either a simple linear extrapolation or the theoretical model-based nonlinear extrapolation. Therefore, a highly extended life expectancy of more than several years can be anticipated for the q-OPV.

Figure 6b. Long-term PCE decay of b- and q-OPVs at 65°C. The PCEs after one year (marked with star) are theoretically predicted values based on the PCE decay as a function of H_{inter} kinetics. The horizontal dashed lines indicate the T_{80} point (defined as the point at which the PCE has fallen to 80% of the value obtained after one day operation)³⁰ for the b- and q-OPVs. The combined experimental and calculated results suggested that the q-OPV after one year would display a PCE of 6.06%, retaining 72.0% of its initial value, whereas the b-OPV would suffer from a much more substantial PCE loss, retaining 44.7% of its initial value.

2. Section "Device Structure and Optical Properties of Materials": How do the author claim a hole rely process under the cascade-energy-level formation? There is no study which proves that except a PL study which reveals the energy transfer from PCDTBT to PTB7. Even the improved IQE can origin from higher charge generation and improved transport. Since the morphology is modified and

also mobility, for sure transport has a dominant role. To claim a cascade charge transfer between the polymers, they should prove it directly by employing for example photo-induced absorption spectroscopy.

Response: We sincerely appreciate the reviewer's comment. As reviewer considered, energy transfer from PCDTBT to PTB7 can be successfully confirmed by our PL analysis, which was also well referenced in previous studies [Adv. Mater. 27, 4398-4404, 2015]. Based on equilibrium thermodynamics, we can assume that there exists the cascade-energy-level between the polymers as well as between the polymers and fullerenes when considering energy band diagram in Figure 1c [Adv. Mater. 27, 4398-4404, 2015]. In particular, recently, Xiao *et al.* suggested that the photo-induced hole transfer from PCDTBT to PTB7 exists in the ternary PTB7:PCDTBT:PC₇₁BM BHJs [Nano Energy 19, 476-485, 2016]. In their study, the steady-state PL data were employed to demonstrate the hole transfer from PC₇₁BM via PCDTBT finally to PTB7. Similarly, Sun *et al.* demonstrated that the small amount of PCDTBT plays the role in the hole relay between PC₇₁BM and PTB7-Th via its energetically adequate HOMO orbital. Indeed, the HOMO energy level of PCDTBT is between those of PTB7-Th and PC₇₀BM, forming cascade HOMO energy levels for more effective extraction of holes from PC₇₁BM [Org. Electron. 37, 222-227, 2016]. As reviewer mentioned, we agreed that it is valuable to directly observe the cascade charge transfer between the components using experimental analysis. Unfortunately, PIA (photo-induced absorption) analysis is not currently available in our experimental circumstances. It should be noted that the main goal of this study is to explore the enhanced long-term morphological stability of the q-BHJ OPVs. Therefore, elucidating transfer mechanism would be beyond the major scope of the present study. Rather, it would be appropriate for next independent study of ours. In this report, we would like to stay more focused on the experimentally obvious results of energy transfer between PCDTBT and PTB7 at equilibrium based on steady-state PL analysis supported by previously reported data of others [Adv. Mater. 27, 4398-4404, 2015 & Nature Commun. 6, 7327, 2015]. A study to investigate the cascade charge transfer between the components is currently underway, and it will be reported in the future. Per reviewer's commentary, we modified the manuscript and Fig. S2 in the revised Supplementary Information to clarify the information and our goal, as below.

→ (Page 5) The photoluminescence (PL) spectra analysis indicates energy transfer between PCDTBT and PTB7 at equilibrium (see Supplementary Note 1 and Supplementary Fig. 2). In particular, the absorption band of PTB7 substantially overlapped the emission band of PCDTBT, which in turn enables the efficient energy transfer from PCDTBT to PTB7^{5,23}. (A further

investigation for the detailed charge transfer mechanism driven by the cascade-energy-level will be reported in the future.)

Supplementary Fig. 2. Transfer process in the blends. PL intensity spectra of pure PCDTBT, pure PTB7, and binary PTB7:PCDTBT blend (The excitation wavelength of 533 nm corresponds to the main absorption region of PCDTBT). Comparison of these spectra provides evidence of energy transfer from PCDTBT to PTB7. The nearly complete PL quenching in the ternary PTB7:PCDTBT:PC₇₁BM blend indicated efficient charge transfer between the polymers and fullerenes, which is a prerequisite for high-performance OPVs⁶.

3. Section "Device Structure and Optical Properties of Materials": What is the motivation of the optical simulation with T-matrix method? Which information are obtained there which is not present in the usual UV-Vis spectroscopy results? I see it a useless piece of info and results in the manuscript which can be simply removed, except they state their motivation and additional achieved information more clearly. The same argument with Gmax compared to the results from simulation.

Response: We appreciate the point raised by the reviewer. We employed the T-matrix method to show quantitative information on the absorption enhancement of the solar spectrum in the q-BHJ OPV. These information would provide a substantial benefit for readers who want to obtain detailed understanding on the mechanism of the OPV. As provided in Supplementary Fig. 3, first of all, T-matrix simulations can reveal detailed information on the distribution of the E-field over the structure along vertical direction as a function of the incident wavelength. In addition, it clearly demonstrates that most of the absorption originates from the blend layer. Typical UV-Vis spectroscopy method mainly concerns the absorption of light in the entire structure not in each of the layers in the structure.

On top of that, a typical UV-Vis spectroscopy is not able to uncover detailed profile of the E-field distribution over each of the layers as a function of incident wavelength. Therefore, a computational method such as T-matrix method can find its usefulness in this point. Subsequently, such information not only highly speaks of the enhanced absorption of the solar spectrum over broad range (300-700 nm) in the q-OPV but also clearly proves that the q-blend is the critical active layer in charge of the enhanced optical properties. Also, the parasitic absorption spectra in Supplementary Fig. 3b were effectively employed to obtain the IQE spectra of devices [$\text{IQE} = \text{EQE}/(1 - R - \text{parasitic absorption})$], as demonstrated in the Methods section. **More importantly, based on the E-field distribution, we can derive the exciton generation rate, which can be applied in the charge carrier transport equation to calculate photovoltaic parameters of the OPV as provided in the Supplementary Note 3.** The G_{max} data was experimentally obtained from photocurrent measurement, and it agreed with the calculation results from the T-matrix method. We would suggest that detailed calculations and information from the modeling rather remain in Supplementary Information for readers. We mentioned in the modified manuscript that the computational approach based on the T-matrix method is sufficiently valid to extract important information on the mechanism and parameters concerning the q-OPV, as provided below.

- (Page 5) The optical simulation results based on the T-matrix method agreed well with the experimental spectra (Supplementary Note 2 and Supplementary Fig. 3a and b).
- (Page 7) The calculated G_{max} agreed with the optically simulated value based on the E-field distribution in the active layer (Supplementary Fig. 3c and d).
- (Page 12) It has been well known that the domain size and its growth rate are strongly associated with the photovoltaic performance⁸⁻¹¹. We therefore further proceeded to numerically obtain the photovoltaic parameters that depended on the H_{inter} , which influences the exciton transport dynamics. To this end, we employed the modified drift-diffusion equation for the charge carriers and the Poisson equation for the electric potential, which used the exciton generation rate as previously provided in Supplementary Note 2 and Supplementary Fig. 3. From the numerical calculations, the time-dependent V_{oc} , J_{sc} , FF, and PCE values can be obtained from the H_{inter} values (details for the procedure and results can be found in Supplementary Note 3 and Supplementary Fig. 18). Interestingly, as Figure 6a indicates, these calculated PCEs well matched the experimentally measured values.
- (Page 16) The IQE spectra were obtained based on $\text{IQE} = \text{EQE}/(1 - R - \text{parasitic absorption})$, where the experimentally obtained spectral EQE and R values were used (refer to parasitic

absorption spectra in Supplementary Fig. 3b as well)⁴⁶.

4. Figure 6 and related discussion: This part is strange to me. The presented data in the first month comes from the experimental results and the rest from the theoretical calculation based on extrapolated H_{inter} ? Why they don't extrapolate just the PCE achieved from experiments and estimate the T80? In figure 6a and SI 17d, there is a slight slope difference between the curves achieved from experiments and calculations. What is the difference of PCE at T80 for these two cases? I guess that this small slope difference will get much more pronounced after long-term calculations based on extrapolation.

Response: We sincerely appreciate thoughtful reviews. More importantly, we take these points to make the revised manuscript deliver clear information. . Therefore, we divided the reviewer's comments into two parts as below and answered separately.

Q4-1) The presented data in the first month comes from the experimental results and the rest from the theoretical calculation based on extrapolated H_{inter} ? Why they don't extrapolate just the PCE achieved from experiments and estimate the T80?

Response: In this manuscript, we thoroughly examined PCEs according to H_{inter} and endeavored to unveil correlation of H_{inter} with PCE. In this sense, we employed a kinetic model for the phase-separated domain governed by spinodal decomposition, which provides a mode for the time-dependent H_{inter} (e.g. $H_{inter}(t) = H_{inter}(t_0) + C(t - t_0)^{1/3}$ relationship in the manuscript.). Based on the numerical calculations of the drift-diffusion model, the time-dependent V_{oc} , J_{sc} , FF, and PCE values can be obtained from the theoretically predicted H_{inter} values at different operation times. Based on a statistically significant fitting between the model and the experimental data for different OPVs, **we could conclude that that the PCE at a long-enough operation time can be successfully estimated using the model.** To secure the reliability of the model, we tried to measure the 4-months (130 days at 65°C) operation data of the q- and b-OPVs in the course of the revision (3 months), and found that the newly measured PCE values stay close on the predicted curves. The long-term stability of the q-OPVs expected from the model over 1-year operation can also be confirmed by comparison of predicted PCEs between a simple linear extrapolation (e.g. 6.04%) vs. the nonlinear H_{inter} -kinetics-based model (e.g. 6.06%). Due to the long-term stability, it is also notable that $T_{80,q}$ either from the simple linear extrapolation or H_{inter} -kinetics-based model show very similar values (e.g. 4.20 vs. 4.36

years). Collectively, we believe that it is reasonable to estimate PCE after one-year operation based on the theoretical model. Per reviewer's comments, we revised manuscript and Figure 6b to clearly provide out motivation of data demonstration, as below.

→ (Page 13) Notably, it is strongly expected that the q-OPV will not reach the $T_{80,q}$ point (defined as the point at which the PCE has fallen to 80% of the value obtained after one day operation)³⁰ even after one year of operation, whereas the b-OPV was estimated to reach the $T_{80,b}$ point within one month. Indeed, we found that $T_{80,q}$ of 4.20 - 4.36 years based on either a simple linear extrapolation or the theoretical model-based nonlinear extrapolation. Therefore, a highly extended life expectancy of more than several years can be anticipated for the q-OPV.

Figure 6b. Long-term PCE decay of b- and q-OPVs at 65°C. The PCEs after one year (marked with star) are theoretically predicted values based on the PCE decay as a function of H_{inter} kinetics. The horizontal dashed lines indicate the T_{80} point (defined as the point at which the PCE has fallen to 80% of the value obtained after one day operation)³⁰ for the b- and q-OPVs. The combined experimental and calculated results suggested that the q-OPV after one year would display a PCE of 6.06%, retaining 72.0% of its initial value, whereas the b-OPV would suffer from a much more substantial PCE loss, retaining 44.7% of its initial value.

Q4-2) In figure 6a and SI 17d, there is a slight slope difference between the curves achieved from experiments and calculations. What is the difference of PCE at T80 for these two cases? I guess that this small slope difference will get much more pronounce after long-term calculations based on extrapolation.

Response: As we discussed in our manuscript, kinetics of H_{inter} can be considered to be dominated by spindodal decomposition, thereby showing $H_{inter}(t) = H_{inter}(t_0) + C(t - t_0)^{1/3}$ relationship. H_{inter} is, thus, not mutated linearly according to time. In this regard, Fig. S18 (e.g. time vs. PCE or H_{inter}) would be more appropriate to deliver information on the time-dependent performance compared to Figure 6a (e.g. H_{inter} vs. PCE]. In Figure 6a, we found that the simple linear connection between the two data points (from 1 day to 30 days) would mislead the readers. We therefore removed the connecting lines as suggested (refer to revised Fig. 6). To secure the statistical reliability, the standard deviation of H_{inter} obtained by AFM imaging of 5 different samples each was also provided in Table S6 in the revised Supplementary Information. Sentences involving the comparison between T_{80} values obtained from experimental and calculated results were newly included in the manuscript, as described in Response for Q4-1. We appreciate for thoughtful commentary from reviewer again.

Figure 6(a). Time-dependent PCE- H_{inter} characteristics of b- and q-OPVs. The calculated PCE values were derived from the time-dependent H_{inter} by using the modified drift-diffusion equation together with the Poisson equation.

Supplementary Fig. 18(d). Time-dependent H_{inter} and H_{inter} -derived PCE.

Supplementary Table 6. Comparison of L_{cor} and H_{inter} of the b- and q-BHJs operated for 1 day and 30 days at 65°C measured by AFM images of 5 different samples each.

	Operation duration	$L_{cor,x}$ (nm)	$L_{cor,y}$ (nm)	H_{inter} (nm)
b-BHJ	1 day	109.9 ± 4.95	250.2 ± 10.26	16.01 ± 0.69
	30 days	144.8 ± 6.67	304.1 ± 14.91	20.82 ± 1.00
q-BHJ	1 day	106.9 ± 3.53	200.3 ± 7.21	15.01 ± 0.53
	30 days	137.4 ± 5.08	151.1 ± 4.68	16.18 ± 0.52

5. SI figure 16: Despite the more stable trend calculated for H_{inter} , PCE shows a faster drop at high temperature for the quaternary system as compared to the binary reference. It is not mentioned and addressed properly in the manuscript.

Response: We apologize for our mistake of insertion of wrong data in the previous version of Fig. S17 in the Supplementary Information. The data points of ‘PCE, b-OPV’ and ‘PCE, q-OPV’ were displayed incorrectly. Reviewer can confirm the correction in Figure 6b and Fig. S18d (e.g. PCE value of b-OPV after one day at 65°C is 5.66% (modified figure), not 4.56% (previous original figure)). We checked the main article, figures, and tables thoroughly and several times for the revision. We apologize again any confusion or misleading and sincerely appreciate reviewer’s comment.

Modified Supplementary F ig. 17. Domain growth analysis with varying operating temperatures.

Comparison of H_{inter} and PCE of b- and q-OPVs as a function of annealing temperature for one day. (Left) Previously depicted wrong data and (right) corrected data by revision.

6. According to literature, multi-component systems, ternaries or quaternaries, may show an adjusted recombination mechanisms compared to their binary reference system. It will influence the lifetime of the solar cells, particularly under light. This important and relevant point should be shortly noticed in the introduction section and properly cited.

Response: As reviewer recommended, we newly mentioned the issue of modified recombination mechanisms in the multi-component BHJs. The sentences and three relevant papers were newly included in Introduction, as shown below.

→ (Page 3) In addition, ternary OPVs provide adjusted recombination mechanisms compared to their binary counterparts, which would prolong the lifetime of OPVs under degradation conditions^{3,6,19}.

***Relevant references**

[3] Lu, L., Xu, T., Chen, W., Landry, E. S. & Yu, L. Ternary blend polymer solar cells with enhanced power conversion efficiency. *Nature Photon.* **8**, 716-722 (2014).

[6] Liu, S. *et al.* Enhanced efficiency of polymer solar cells by adding a high-mobility conjugated polymer. *Energy Environ. Sci.* **8**, 1463-1470 (2015).

[19] Zhang, Y. *et al.* Synergistic effect of polymer and small molecules for high-performance ternary organic solar cells. *Adv. Mater.* **27**, 1071-1076 (2015).

7. Table 5 in SI: A comparison between the achieved results under dark condition to those results obtained under light, as well as encapsulated and not-encapsulated devices is really inaccurate and unfair comparison.

Response: In Supplementary Table 7 (previous Supplementary Table 5), we intended to include reported results to justify importance of the stability study in PV community. From reviewer's suggestion, we newly included the results of photo-induced degradation under AM 1.5G illumination, as provided in Fig. S19 and Table S7 in the revised Supplementary Information. **For real outdoor application approach, the devices completed with encapsulation were illuminated under AM 1.5G solar simulator with a 12 h light/dark illumination cycle for ~21 days.** The q-OPV retained more than 61.16% of its initial PCE after ~21 days of operation under illumination (5.15% in the PCE), while the b-OPV maintained only 24.76% of its initial PCE at the same conditions (1.88% in the PCE). It is notable that our q-OPV exhibited a higher PCE in comparison with the previously reported state-of-the-art OPVs under the photo-induced degradation condition (Supplementary Table 7). Collectively, the long-term stability of the q-OPV under thermal and photo-induced degradation conditions strongly suggests that the quaternary BHJ systems are advantageous to outdoor application fields. We believe that our report can inspire stability issue of OPVs as well as provide a guideline toward high-efficiency OPVs with long-term stability. We, in this study, aimed to gain insight into the enhanced long-term morphological stability in quaternary BHJs. Therefore, the main degradation factor was selectively limited to thermal stress, while other degradation parameters (*e.g.*, water, oxygen, light soaking, or UV stress) were minimized by encapsulating the devices and then aging them in the dark. We modified the relevant sentences and Supplementary Table 7, and Supplementary Fig. 19 was newly included in the manuscript, as below.

→ (Page 13) For real outdoor application approach, long-term stability of the devices was further explored under light exposure (1 Sun condition). As provided in Supplementary Fig. 19, our q-OPV exhibited significantly better performance than that of the b-OPV under the photo-induced degradation conditions (5.15% vs. 1.88% in the PCE after ~21 days). Also, it is noteworthy that our q-OPV exhibited higher PCE throughout the duration of the thermal treatment or light illumination compared to literature values (see Supplementary Table 7). The long-term stability of the q-OPV and its superior photovoltaic performances strongly suggest that this OPV can be used in outdoor applications with commercially acceptable quality.

Supplementary Fig. 19. Long-term photo-induced degradation test. Long-term PCE decay of b- and q-OPVs under AM 1.5G solar simulator with a 12 h light/dark illumination cycle for ~21 days.

Supplementary Table 7. Superior PCE sustainability of our q-OPV. Comparison of the PCE decay (the percentage of PCE decrease relative to the initial value) between our q-OPV and other state-of-the-art binary OPVs under diverse aging conditions (some PCE values estimated from the figure images, and not exactly stated in literature, are labeled with *ca*).

D:A material	Device structure	Initial PCE	Last PCE	PCE loss	Degradation conditions (e.g., light, temperature, encapsulation)	Ref.
q-OPV	Inverted	8.42%	6.27% after 30 days (experimental) & 6.06% after one year (simulated)	25.53% & 28.03%	Dark, 65°C, with encapsulation	This study
b-OPV		7.59%	4.56% after 30 days (experimental) & 3.39% after one year (simulated)	39.92% & 55.34%		
q-OPV		8.42%	5.15% after ~21 days	38.84%		
b-OPV		7.59%	1.88% after ~21 days	75.23%	Illumination under AM 1.5G solar simulator (12 h light/dark illumination cycle), with encapsulation	
P3HT:PC ₆₁ BM	Standard	3.0%	ca. 1.5% after 4700h	ca. 50%	Continuous illumination under a Sulphur plasma lamp, 50°C (testing chamber temperature), with encapsulation	18
P3HT:PC ₆₁ BM	Standard	3.7%	2.5% after 1000h	32.43%	Dark, 45°C, w/o encapsulation (inert measurement conditions)	19

P3HT:PC ₆₁ BM	Standard	3.2%	1.8% after 1000h	43.75%	Continuous illumination under a 150W Xenon lamp with AM 1.5G filter, 45°C, w/o encapsulation (inert measurement conditions)	19
P3HT:PC ₆₁ BM	Standard	4.0 ± 0.05%	ca. 2.92% after 4400 h (experimental) & ca. 2.72% after 3.1 years (simulated)	27% & 32%	Continuous illumination under a Sulfur plasma lamp (6000 K), 37°C, with encapsulation	20
PCDTBT:PC ₇₁ BM	Standard	5.5 ± 0.15%	ca. 3.74% after 4400h (experimental) & ca. 3.19% after 6.2 years (simulated)	32% & 42%		
PCDTBT:PC ₇₁ BM	Standard	7.04%	ca. 5.56% after 19500h (simulated) & ca. 5.63% after 650 days (experimental)	21.02% & 20.03%	Continuous illumination under a Sulphur plasma lamp (6000 K), room temperature, with encapsulation	21
PCDTBT:PC ₇₁ BM	Standard	6.50%	ca. 3.25% after 30 days	ca. 50%	Under ambient air conditions, w/o encapsulation	22
PCDTBT:PC ₇₁ BM	Standard	5.02%	3.54% after 4500h	29.48%	Continuous illumination under a halide lamp (1000 Wm ⁻²), 45°C, with encapsulation	23
PTB7:PC ₇₁ BM	Inverted	5.37%	ca. 3.33% after ca. 3500h	ca. 37.98%	Under ambient dark conditions, w/o encapsulation	24

Reviewer #3 (Remarks to the Author):

Summary:

In recent years, ternary blend organic solar cells have been presented as a viable means of extending the light absorbance in organic solar cells. This study presents a quaternary blend solar cell with improved performance over the binary or ternary blend cells based on its components. This is due in large part to the extended light absorption and balanced carrier mobilities, as shown in UV-Vis, quantum efficiency measurements, and SCLC measurements. More importantly, the authors show that the quaternary blend cell has enhanced stability over the binary and ternary blends, maintaining 72% of the original performance after significant thermal aging. GIWAX studies, AFM, and simulations of domain growth indicate that this stability is due to reduced domain growth in the quaternary blend. Using multiple components to control the crystallization etc. is a novel idea.

This paper presents a thorough study of a quaternary blend organic solar cell. The improved performance over the binary and ternary blends is interesting in and of itself, but coupled with the enhanced stability, this represents an intriguing result. The reviewer recommends this paper for acceptance with the following minor revisions.

Overall response: We authors appreciate valuable commentary and positive decision from the reviewer. We have concentrated on securing the stability results of the q-OPV for sufficiently long duration. Given the maximum period of time for the revision (3 months), we could extend the device stability test by measuring it at the point of 4 months operation (1 + 3 months, ~130 days). From the updated test, we confirmed that the long-term operation stability results agreed well again with our theoretical model (refer to modified Figure 6 b). Based on the updated results and the previous analysis, we could secure strongly conclusive evidence and more profound understanding on the stability enhancement of the proposed q-OPVs.

Minor Issues

1. When discussing the energy transfer between donor materials, the authors assert that the "combination of two donors was also beneficial for the energy transfer kinetics as revealed in photoluminescence." While, it is clear that energy or charge transfer is occurring between the two polymers, it is not clear that this aids the quenching as compared to the binary blends. To make this assertion, we would need to compare the PL of the binary blends. Additionally, the presented PL measurements are steady state, and do not explain anything about the kinetics of the systems.

Response: We sincerely appreciate for reviewer's comments and recommendations. As reviewer considered, energy transfer from PCDTBT to PTB7 was successfully examined by our PL analysis. The main purpose of the PL analysis was to confirm the energy transfer from PCDTBT to PTB7, as previously demonstrated in literature [Adv. Mater. 27, 4398-4404, 2015]. Besides, the PL quenching in the ternary PTB7:PCDTBT:PC₇₁BM BHJs insinuated efficient charge transfer between the polymers and fullerenes [Nano Energy 19, 476-485, 2016 & Org. Electron. 37, 222-227, 2016]. Indeed, our finding of nearly complete quenching in the ternary blend indicates highly effective charge transfer in the multi-component BHJ systems, which is a prerequisite for high-performance OPVs. It is very important to define cascade charge transfer between the components via thorough analysis, as reviewer's consideration. Unfortunately, PIA (photo-induced absorption) analysis is not currently available in our experimental circumstances. It should be noted that the main goal of this study is to explore the enhanced long-term morphological stability of the q-BHJ OPVs. Therefore, elucidating transfer mechanism would be beyond the major scope of the present study. Rather, it would be appropriate for next independent study of ours. In this report, we would like to stay more focused on the experimentally obvious results of energy transfer between PCDTBT and PTB7 at equilibrium based on steady-state PL analysis supported by previously reported data of others [Adv. Mater. 27, 4398-4404, 2015 & Nature Commun. 6, 7327, 2015]. A study to investigate the cascade charge transfer between the components is currently underway, and it will be reported in the future. Per reviewer's commentary, we modified the manuscript and Fig. S2 in the revised Supplementary Information to clarify the information and our goal, as below.

→ (Page 5) The photoluminescence (PL) spectra analysis indicates energy transfer between PCDTBT and PTB7 at equilibrium (see Supplementary Note 1 and Supplementary Fig. 2). In particular, the absorption band of PTB7 substantially overlapped the emission band of PCDTBT, which in turn enables the efficient energy transfer from PCDTBT to PTB7^{5,23}. A further investigation for the detailed charge transfer mechanism driven by the cascade-energy-level will be reported in the future.

Supplementary Note 1. Transfer process in the blends

As shown in Supplementary Figure 2, the emission intensity from PCDTBT (centered at 705 nm) decreased, while the PTB7 emission (centered at 770 nm) increased when PCDTBT was mixed with PTB7 such that PTB7:PCDTBT = 0.5:0.5. This PL emission change strongly indicates the effective

energy transfer from PCDTBT to PTB7^{4,5}. On the other hand, when fullerene was mixed in the PTB7:PCDTBT blend (e.g., PTB7:PCDTBT:PC₇₁BM = 0.5:0.5:1.5), the PL emission peaks from both PCDTBT and PTB7 were distinctively quenched. This can be derived by efficient charge transport between polymers and fullerenes (refer to energy band diagram in Figure 1c)⁶. Therefore, the operation of the q-OPV relies on both energy and charge transfer among the donors and acceptors in the BHJ.

Supplementary Fig. 2. Transfer process in the blends. PL intensity spectra of pure PCDTBT, pure PTB7, and binary PTB7:PCDTBT blend (The excitation wavelength of 533 nm corresponds to the main absorption region of PCDTBT). Comparison of these spectra provides evidence of energy transfer from PCDTBT to PTB7. The nearly complete PL quenching in the ternary PTB7:PCDTBT:PC₇₁BM blend indicated efficient charge transfer between the polymers and fullerenes, which is a prerequisite for high-performance OPVs⁶.

2. In Figure 2.e, it is not clear what the "EQE Enhancement" is, or how it was calculated. Is this the enhancement compared to the binary blends? If so, which one?

Response: The spectral EQE enhancement was obtained from the ratio between quaternary (or ternary) device and PTB7-based binary device. As reviewer pointed out, we modified Figure 2(e) caption as below.

➔ (Page 24) **Figure 2. Optimization of compositions of the donors and acceptors in the q-OPV.** Photovoltaic parameters as a function of PCDTBT concentration (x , $0 \leq x \leq 1$) in PTB7:PCDTBT:PC₇₁BM ($1-x:x:1.5$) blends (a and b) and PC₆₁BM concentration (y , $0 \leq y \leq 1.5$) in PTB7:PCDTBT:PC₇₁BM:PC₆₁BM ($0.9:0.1:1.5-y:y$) blends (c and d). The mean values were

calculated using data from more than 16 cells. (e) EQE spectra and (f) $J_{ph}-V_{eff}$ characteristics of the b-, t-, and q-OPVs. The EQE enhancement indicates the ratio of ternary (dotted turquoise line) or quaternary (dotted red line) devices to the PTB7-based binary device.

3. Ln 196, The authors claim that the q-OPV shows improved carrier transport after thermal treatment. In reality, the q-OPV shows less absolute improvement in the carrier transport than the b-OPV or t-OPV, but this is indicative of balanced charge transport and ideal domain size. This should be clarified.

Response: We sincerely appreciate for your advice. The relevant sentences have been modified as below.

→ (Page 9) Moreover, the quaternary device exhibited relatively balanced mobilities during the one day thermal treatment, also indicative of balanced charge transport and ideal domain size (Supplementary Fig. 6b)²⁷.

4. While the paper is certainly readable, there are several phrases which are awkward or unclear, and should be edited grammatically. The manuscript would be more readable if it can go another round of professional proof-reading. A few examples:

- Ln 50: "In this standpoint" should be "From this standpoint", and the following sentence is unclear.
- Ln 93: "Accorded well" should be "agreed well" or something similar
- Ln 164-167: "temperature exhibited that the performance" is awkward, and the sentence is unclear in general
- Ln 235-236: "A very long time of use" and "Maintained quite well its performance"

Response: Thank you for the thoughtful review and comments. We revised several phrases according to reviewer's suggestion in order to improve reader's understanding. We believe the manuscript has been improved as a result. The revised sentences in the main article are as below.

Comment 4-1) Ln 50: "In this standpoint" should be "From this standpoint", and the following sentence is unclear.

→ (Page 3) From this standpoint, employing an appropriate D or A additive as a morphology stabilizer would kinetically arrest the morphology at its optimum, enabled by providing parameters to control blending/separation behaviors of components¹⁵.

Comment 4-2) Ln 93: "Accorded well" should be "agreed well" or something similar.

→ (Page 5) The optical simulation results based on the T-matrix method agreed well with the experimental spectra (Supplementary Note 2 and Supplementary Fig. 3a and b).

Comment 4-3) Ln 164-167: "temperature exhibited that the performance" is awkward, and the sentence is unclear in general.

→ (Page 8) Interestingly, a thermal-dependent property showed that the performance decay of the b-OPV was accelerated at around 65°C while the q-OPV displayed a better resistance to the decay even at elevated temperatures over 65°C (Supplementary Fig. 9).

Comment 4-4) Ln 235-236: "A very long time of use" and "Maintained quite well its performance".

→ (Page 11) As a consequence of those analyses, the q-OPV was found to be advantageous in retaining a high PCE for the extremely extendable operation duration (Figure 5m). As the figure indicates, the q-OPV exhibited a strong resistance to the performance reduction even after a one-month operation at 65°C (e.g., > 95% of the reference PCE).

REVIEWERS' COMMENTS:

Reviewer #2 (Remarks to the Author):

Authors have addressed my concerns properly. I would recommend publishing this manuscript as it is in journal of Nature Communication.

Reviewer #3 (Remarks to the Author):

In recent years, ternary blend organic solar cells have been presented as a viable means of extending the light absorbance in organic solar cells. This study presents a quaternary blend solar cell with improved performance over the binary or ternary blend cells based on its components. This is due in large part to the extended light absorption and balanced carrier mobilities, as shown in UV-Vis, quantum efficiency measurements, and SCLC measurements. Additionally, the authors show that the quaternary blend cell has enhanced stability over the binary and ternary blends, maintaining 72% of the original performance after significant thermal aging. GIWAX studies, AFM, and simulations of domain growth indicate that this stability is due to reduced domain growth in the quaternary blend.

This paper presents a thorough study of a quaternary blend organic solar cell. The improved performance over the binary and ternary blends is interesting in and of itself, but coupled with the enhanced stability, this represents an intriguing result.

The issues raised by this reviewer have been dealt with by sufficient changes to the manuscript. This paper is recommended for publication.

[Note from the editor: Referee #3 was asked to check if the concerns of referee #1 were addressed, which was confirmed in confidential comments to the editor]